# Bioactive Hydrogel Formulation Based on Ferulic Acid-Grafted Nano-Chitosan and Bacterial Nanocellulose Enriched with Selenium Nanoparticles from Kombucha Fermentation

**DOI:** 10.3390/jfb15070202

**Published:** 2024-07-22

**Authors:** Naomi Tritean, Luminița Dimitriu, Ștefan-Ovidiu Dima, Marius Ghiurea, Bogdan Trică, Cristian-Andi Nicolae, Ionuț Moraru, Alina Nicolescu, Anisoara Cimpean, Florin Oancea, Diana Constantinescu-Aruxandei

**Affiliations:** 1Bioresource and Polymer Department, National Institute for Research & Development in Chemistry and Petrochemistry—ICECHIM, Splaiul Independenței nr. 202, Sector 6, 060021 Bucharest, Romania; naomi.tritean@icechim.ro (N.T.); luminita.dimitriu@icechim.ro (L.D.); ovidiu.dima@icechim.ro (Ș.-O.D.); marius.ghiurea@icechim.ro (M.G.); bogdan.trica@icechim.ro (B.T.); cristian.nicolae@icechim.ro (C.-A.N.); 2Faculty of Biology, University of Bucharest, Spl. Independentei nr. 91-95, Sector 5, 50095 Bucharest, Romania; anisoara.cimpean@bio.unibuc.ro; 3Laboratoarele Medica Srl., Frasinului Str. nr. 11, 075100 Otopeni, Romania; ionutmoraru@pro-natura.ro; 4“Petru Poni” Institute for Macromolecular Chemistry, Aleea Grigore Ghica Voda 41A, 700487 Iasi, Romania; alina@icmpp.ro

**Keywords:** Kombucha nanocellulose, biopolymer-based hydrogel, nanoformulation, titanium surface adhesion, antimicrobial, cytocompatibility, antioxidant, anti-inflammatory, gingival fibroblasts

## Abstract

Selenium nanoparticles (SeNPs) have specific properties that result from their biosynthesis particularities. Chitosan can prevent pathogenic biofilm development. A wide palette of bacterial nanocellulose (BNC) biological and physical-chemical properties are known. The aim of this study was to develop a hydrogel formulation (SeBNCSFa) based on ferulic acid-grafted chitosan and bacterial nanocellulose (BNC) enriched with SeNPs from Kombucha fermentation (SeNPsK), which could be used as an adjuvant for oral implant integration and other applications. The grafted chitosan and SeBNCSFa were characterized by biochemical and physical-chemical methods. The cell viability and proliferation of HGF-1 gingival fibroblasts were investigated, as well as their in vitro antioxidant activity. The inflammatory response was determined by enzyme-linked immunosorbent assay (ELISA) of the proinflammatory mediators (IL-6, TNF-α, and IL-1β) in cell culture medium. Likewise, the amount of nitric oxide released was measured by the Griess reaction. The antimicrobial activity was also investigated. The grafting degree with ferulic acid was approximately 1.780 ± 0.07% of the total chitosan monomeric units, assuming single-site grafting per monomer. Fourier-transform infrared spectroscopy evidenced a convolution of BNC and grafted chitosan spectra, and X-ray diffraction analysis highlighted an amorphous rearrangement of the diffraction patterns, suggesting multiple interactions. The hydrogel showed a high degree of cytocompatibility, and enhanced antioxidant, anti-inflammatory, and antimicrobial potentials.

## 1. Introduction

The etiology of periodontal disease is both complex and multifactorial [1]. The inflammation starts with changes in the ratio of beneficial and pathogenic bacteria (host-microbiome imbalance). The failure of inflammation resolution will trigger hyper-inflammatory responses and pathogenic bacteria proliferation, which will tip the balance from gingivitis to periodontitis, the latter leading to loss of connective tissue attachment and bone resorption [2,3]. When there is no effective treatment in order to cure these pathologies, dental implants will eventually be used as a substitute for the natural root of the tooth. Hydrogel (nano)formulations based on natural biopolymers have a great potential to enhance several dental implant outcomes by preventing infection development, reducing inflammation, and improving the integration of the implant with the surrounding tissues [4,5,6]. Also, the embedding of other promising biogenic nanostructures (e.g., nanoparticles) may lead to increased beneficial effects [7,8].

Nanobiotechnology has an interdisciplinary approach by combining the fundamentals and techniques from various scientific fields, leading to novel breakthroughs in nanoscale applications [9]. It has a great potential in biomedicine, especially due to the specific biological activity of several nanomaterials, which results from their particular physical-chemical properties achieved during their (bio)synthesis process [10,11,12].

Never-dried bacterial nanocellulose (NDBNC) and (nano)chitosan (CS) are natural, biocompatible, and biodegradable (nano)biopolymers that can closely mimic the structural and biochemical properties of the fibrillar native extracellular matrix (ECM), providing a favorable microenvironment for cell attachment, proliferation, and differentiation [13,14,15]. Chitosan nanostructures target dysbiotic biofilm, with CS already proven to prevent dysbiotic biofilm establishment and promote biofilm dispersal [16,17,18,19]. NDBNC can also significantly improve the mechanical properties of CS-based hydrogels due to its outstanding features like high mechanical strength, high specific surface area, chemical stability, high surface chemistry, hydrophilicity, crystallinity, and transparency [20,21].

Moreover, the physical, chemical, and biological properties of the NDBNC-CS matrix can be enhanced by embedding other bioactive compounds. Hydroxycinnamic acids (HCA) have been shown to have various biological activities that are relevant for the biomedical field. They can also act on dysbiotic biofilm and oxidative stress [22,23]. However, due to their rather low solubility in water, they have limited bioavailability, which decreases their full potential activity. The grafting of HCA to CS leads to an HCA-CS matrix with increased water solubility. As a result, the biological activity is also improved [22,24,25,26,27], mainly due to increased bioavailability. Ferulic acid-grafted chitosan was proven to be the most effective against several Gram-positive and Gram-negative bacteria, including methicillin-resistant *Staphylococcus aureus* strains, in comparison with other HCA-CS conjugates, i.e., caffeic acid-grafted chitosan and sinapic acid-grafted chitosan [28].

Nanoparticles (NPs) are key nanostructures used in biomedicine as drug carriers, for diagnostic and/or theragnostic, for tissue regeneration, and in 3D cell culture due to their unique size, shape, chemical composition and stability in solution [29]. Selenium nanoparticles (SeNPs) have tremendous potential in terms of biological activity if the dose is optimal. It is well known that there is a strong relationship between Se and human health, which is one of the major advantages of SeNPs in comparison to other nanoparticles [30,31]. According to the European Food Safety Authority, an adequate intake (AI) of 70 µg/day for adults was established. In lactating women, the AI was set at 85 µg/day in order to replace the Se secreted in breast milk [31]. A number of health disorders have been linked to Se deficiency, e.g., brain, liver, thyroid, and reproductive disfunctions. Se is involved in expression and proper function of selenoproteins, provides protection against oxidative stress, and is essential for cellular function in general. Therefore, Se possesses both prophylactic and therapeutic effects [30]. Selenium has a rather narrow range when it comes to the shift from the beneficial to the toxic dose, but the toxicity is reduced in the case of SeNPs compared to other selenium forms [32,33,34]. Moreover, green, environmentally friendly methods for the biosynthesis of SeNPs, such as microbiological biosynthesis or plant extracts, were proven to reduce further SeNP toxicity due to the capping biocorona formed from biomolecules. These approaches confer a unique character to SeNPs through the nature of their biocorona, which additionally increases SeNP stability and bioavailability. SeNPs exhibit significant antioxidant activity, which protect cells from oxidative stress [35,36], as well as increased antimicrobial properties against several pathogenic bacteria [37,38,39], pathogenic yeast [40,41], and fungi [42,43]. Additionally, SeNPs have been proven to promote inflammation resolution [44,45]. In our previous work, we demonstrated the bioactivity of selenium nanoparticles from Kombucha fermentation [46], as well as the Iβ-cellulose-reinforced structure of BNC by Kombucha fermentation with pollen and selenium. Moreover, we investigated in a previous study the biological activity of a hydrogel nanoformulation with a CS-BNC matrix embedding SeNPs phytosynthesized from sea buckthorn leaf extract [19]. The aim of this study was focused on the development of a hydrogel formulation based on ferulic acid-grafted chitosan and NDBNC enriched with SeNPs from Kombucha fermentation (SeNPsK) with increased cytocompatibility, antioxidant, and antibacterial activities compared with the previous hydrogel, as well as a high anti-inflammatory effect.

## 2. Materials and Methods

### 2.1. Materials

The investigation of cytocompatibility, antioxidant, and anti-inflammatory potential was conducted on HGF-1 human gingival fibroblasts (CRL-2014, American Type Cell Culture Collection). *Limosilactobacillus reuteri* strain DSM 20016 and *Ligilactobacillus salivarius* strain DSM 20555 were used in order to investigate the potential of the hydrogel formulations to stimulate probiotic growth. The microbial strains used for assessing the antimicrobial activity were: *S. aureus* ATCC 25923, *Bacillus cereus* NCTC 10320, *Pseudomonas aeruginosa* ATCC 27853, *Escherichia coli* ATCC 25922, and *Candida albicans* ATCC 10231.

The following culture media were used: Müeller–Hinton agar (MHA) and Müeller–Hinton broth (MHB) (Scharlau, Barcelona, Spain) for pathogen bacterial growth; Sabouraud dextrose agar (SDA) and Sabouraud dextrose broth (SDB) for yeast growth (Scharlau, Barcelona, Spain); De Man-Rogosa-Sharpe agar (MRSA); and De Man-Rogosa-Sharpe browth (MRSB) (Scharlau, Barcelona, Spain) for lactobacilli growth. The HGF-1 cells were maintained in Dulbecco’s Modified Eagle’s Medium-low glucose (DMEM) (Sigma-Aldrich, St. Louis, MO, USA) supplemented with fetal bovine serum (FBS) USDA APPD. ORIGIN (Thermo Fisher Scientific, Waltham, MA, USA), sodium bicarbonate, and D-(+) antibiotic antimycotic solution 100× stabilized (Sigma-Aldrich, St. Louis, MO, USA).

Cell viability and proliferation were assessed using cell counting kit-8 (Bimake, Houston, TX, USA), Viability/Cytotoxicity Assay Kit (Biotium, Fremont, CA, USA). Phalloidin-iFluor 488Reagent (Abcam, Cambridge, United Kingdom) was used for actin filament labeling, and the nuclei were marked with 4′,6-diamidino-2-phenyindole, dilactate (DAPI) (Sigma-Aldrich, St. Louis, MO, USA). The in vitro antioxidant activity was investigated using 2,7-dichlorodihydrofluorescein (Cayman Chemicals, Ann Arbor, MI, USA) in the presence of 30% hydrogen peroxide (Scharlau, Barcelona, Spain), while the inflammatory response was determined by the LEGEND MAX Human TNF-α ELISA Kit, LEGEND MAX Human IL-6 ELISA Kit, and LEGEND MAX Human IL-1β ELISA Kit (BioLegend, San Diego, CA, USA). The nitric oxide production was investigated using the Griess reagent system (Promega Corporation, Madison, WI, USA) in the presence of lipopolysaccharide from *Porphyromonas gingivalis*, lipopolysaccharide from *Escherichia coli* (Sigma-Aldrich, St. Louis, MO, USA).

Ferulic acid-grafted chitosan was prepared using the following reagents: chitosan (10–120 cps) of fungal origin (Glentham Life Sciences, Wiltshire, UK), ascorbic acid, hydrogen peroxide 30% (Scharlau, Barcelona, Spain), ferulic acid (Sigma-Aldrich, St. Louis, MO, USA), absolute ethanol 99.5%, and acetic acid (Chimopar Srl., Bucharest, Romania).

The total polyphenol content, quantification of amino groups and antioxidant activities were assessed using the following chemicals: Trolox 97% (Acros Organics, Thermo Fisher Scientific, Pittburghs, PA, USA), 2,2-diphenyl-1-picrylhydrazyl (Sigma-Aldrich, St. Louis, MO, USA), hydrochloric acid 37% (Chimopar Srl., Bucharest, Romania), Folin–Ciocalteu’s phenol reagent, sodium carbonate, iron chloride (III) (Merck, Darmstadt, Germany), 2,4,6-tri (2-pyridyl-1,3,5-triazine) 98% (Janssen Chimica, Beerse, Belgium)m D-(+)-glucosamine hydrochloride cell culture (MP Biomedicals, LLC, Illikirch, France), ninhydrin (Riedel-de Haën, Seelze, Germany), sodium acetate anhydrous, Dimethyl sulfoxide (DMSO) (Scharlau, Barcelona, Spain).

Other reagents used for various buffers were: trypsin from porcine pancreas, sodium deoxycholate, di-sodium hydrogen phosphate dihydrate, sodium dihydrogen phosphate monohydrate, tris-(hydroxymethyl)-aminomethane, potassium chloride, sodium chloride, paraformaldehyde, Triton X-100, sodium n-dodecyl sulfate 99%, and albumin bovine fraction V, pH 7.0 (Janssen Chimica, Beerse, Belgium).

### 2.2. Preparation of Ferulic Acid-Grafted Chitosan

The intrinsic viscosity and molecular weight of CS were determined with a viscometer Schott AVS 350, CT 62 (Schott-instruments, Mainz, Germany) and a capillary no. 0, based on the determination of the flow time of each solution and solvent at 25 °C. First, from an initial 0.5 g/dL chitosan stock solution, 5 solutions with different concentrations (0.02; 0.04; 0.06; and 0.08 si 0.1 g/dL) were prepared in an acetic acid (0.3 M)/sodium acetate (0.2 M) solution. The relative viscosity (η*_rel_*), specific viscosity (η*_sp_*), and reduced viscosity (η*_red_*) were determined according to Equations (1), (2), and (3), respectively:(1)$$\eta_{rel}=\frac{t_{i}}{t_{0}}$$
(2)$$\eta_{sp}=\eta_{rel}-1$$
(3)$$\eta_{red}=\frac{\eta_{sp}}{c_{sol}}$$
where *t_i_* = solvent flow time; *t*_0_ = flow time of chitosan solutions; *c_sol_* = the concentration of chitosan solutions correlated with η*_sp_* values.

The reduced viscosity was plotted against the concentrations of chitosan solutions tested, and the intrinsic viscosity [*η*] was determined as the intercept of the linear fit.

The molecular weight was determined based on Mark–Houwink–Sakurada Equation (4), where [*η*] is the intrinsic viscosity, *M_w_* is the molecular weight of the chitosan, *k* = 7.4 × 10^−4^ g/dL, and *a* = 0.76:(4)$$\left[\eta \right]=k\times M_{w}^{a}$$

The deacetylation degree of CS, CSFa(−), and CSFa(+) was determined based on the ^1^H NMR spectra recorded on a Bruker Avance NEO 600 MHz spectrometer equipped with a 5 mm inverse detection z-gradient probe at 60 °C. The chitosan solution in slightly acidic, deuterated water was prepared as follows: 10 mg chitosan, 0.8 mL D_2_O, and 10 µL HCl 37% to completely solubilize the chitosan. The deacetylation degree (*DD*) was calculated based on Equation (5) [47]:(5)$$DD\left(\%\right)=\left(1-\frac{1/3\times I_{CH_{3}}}{1/6\times I_{H_{2}-H_{6}}}\right)$$
where *DD* is the deacetylation degree, *I*_CH3_ is the signal intensity of the -CH_3_ protons from the acetyls of chitosan (2.06 ppm), and *I*_H2–H6_ is the signal intensity of chitosan protons from C2 to C6 of the d-glucopyranosyl ring (3–4.2 ppm). The intensity was calculated by numeric integration of the ^1^H NMR signals, as shown in Appendix A.

Ferulic acid-grafted chitosan, denoted CSFa(+), was synthesized according to a previous study [28], which involved the following steps: dissolving 2 g of chitosan (CS) in 180 mL of 2% (*v*/*v*) acetic acid solution under continuous stirring; slowly adding 4 mL of 1 M H_2_O_2_ containing 0.432 g of ascorbic acid (AscA), drop by drop. After 30 min of reaction, a 6.2 mM ferulic acid (Fa) solution prepared in 96% ethanol was gradually added to the mixture in a molar ratio of chitosan monomer unit:Fa of 1:0.1 (considering both acetylated and non-acetylated monomer units, i.e., the average molecular weight of monomer unit), and the resulting mixture was maintained at room temperature for 24 h with continuous stirring. The obtained grafted chitosan solution was dialyzed in double-distilled water at room temperature for 72 h using a regenerated cellulose (RC) dialysis membrane (28 mm diameter) with a molecular weight cut-off (MWCO) of 3500 Da. In order to remove the unreacted FA, the double-distilled water was changed 3 times/day, until no free ferulic acid was detected using the Folin–Ciocalteu reagent. The non-grafted chitosan, denoted CSFa(−), was prepared in the same way as CSFa(+), except that no Fa was added. The samples were freeze-dried for 72 h using a ScanVac CoolSafe 55-4 freeze-dryer (LaboGene, Bjarkesvej, Denmark) at a −55 °C freeze-drying temperature. After the freeze-drying process, the samples were kept in the desiccator and analyzed in less than 24 h for all the physical-chemical analysis. The water content of the samples after the freeze-drying process was considered structural water or crystallization water.

#### 2.2.1. Investigation of Grafting Degree

The quantification of grafting degree was determined with two methods: Folin–Ciocalteu and ninhydrin reagent, respectively.

Folin–Ciocalteu reagent: Over 50 μL of 10 mg/mL (*w*/*v*) sample, 450 μL of double-distilled water, and 50 μL of Folin–Ciocalteu reagent were added, shaking the tubes for 5 min. Subsequently, 500 μL of 7% Na_2_CO_3_ and 200 μL of double-distilled water were added. After 60 min of incubation at room temperature, the mixtures were centrifuged at 6000 rcf, and the absorbance values were recorded at λ = 765 nm (CLARIOstar BMG Labtech, Ortenberg, Germany). The calibration curve was carried out starting from a stock solution of 500 μg/mL FA in 70% ethanol. The amount of grafted FA was expressed as mg FA/g chitosan. The molar ratio between FA and chitosan was calculated based on their molecular weights. The grafting degree was expressed as the percent of grafted chitosan monomer units and was determined according to the following Equation (6):(6)$$\mathrm{Grafting}\;\mathrm{degree}\;\left(\%\right)=\frac{N_{gm}\;}{N_{m}}\times 100$$
where *N_gm_* represents the number of grafted chitosan monomer units and was calculated as grafted FA-equivalent moles/g chitosan, and *N_m_* represents the average moles of chitosan monomer units in a gram of grafted chitosan.

Ninhydrin reagent: The amino groups in CS, CSFa(−), and CSFa(+) were quantified by ninhydrin assay according to [48] with some modifications. Briefly, 0.2 mL of chitosan samples in 2% acid acetic were mixed with 0.2 mL of 1 M acetate buffer pH = 5 and 0.6 mL of ninhydrin reagent prepared by solubilization of 10 mg/mL ninhydrin in dimethyl sulfoxide (DMSO). The samples were incubated at 95 °C for 15 min. in a digital dry bath (BSH1004, Benchmark Scientific, Sayreville, NJ, USA). After cooling, the absorbance spectra were recorded using a microplate reader. The free amino groups were quantified based on the absorbance at 580 nm against a D-glucosamine standard curve prepared from a stock solution of 0.4 mg/mL.

A qualitative analysis of the grafting was performed using UV–Vis spectroscopy by recording the spectra of 10 µg/mL AscA in double-distilled water (ddH_2_O), 16 µg/mL Fa in absolute ethanol, 400 µg/mL CS, CSFa(−), and, respectively, CSFa(+) in 2% acetic acid with a microplate reader.

#### 2.2.2. Dynamic Light Scattering (DLS) and Zeta Potential Analysis

DLS analysis and zeta potential of CS, CSFa(−), and CSFa(+) in a 2% acetic acid solution, as well as in double-distilled water for CSFa(−) and CSFa(+), were assessed using the AMERIGO™—particle size and zeta potential analyzer (Cordouan Technologies, Pessac, France), according to the manufacturer’s instructions. The AmeriQ software version 3.2.3.0 was used in order to analyze the results.

### 2.3. Preparation of Hydrogel Enriched with Selenium Nanoparticles from Kombucha Fermentation

The SeNPsK-free hydrogel (BNCSFa) was prepared by adding 7% (*w*/*v*) CSFa(+) in a suspension of 0.4% (*w*/*v*) NDBNC derived from Kombucha fermentation with pollen and prepared in double-distilled water. The mixture was homogenized using an Ultra-Turrax homogenizer (IKA, Staufen, Germany). The NDBNC was characterized in a previous study [46], and it was obtained after the washing, milling, and 20 microfluidization passes (LM20 microfluidizer, Microfluidics, Westwood, MA, USA) of the bacterial cellulose from the Kombucha fermentation with 25 g polyfloral pollen, 190 mg sodium selenite, and 30 mL of Symbiotic Culture of Bacteria and Yeast (SCOBY)/300 mL sweetened black tea.

For the physical-chemical characterization, the hydrogel with SeNPsK (SeBNCSFa) was prepared by adding 2.5 µg/mL of SeNPsK in a suspension of 0.4% (*w*/*v*) NDBNC. After vigorous mixing, 7% ferulic acid-grafted chitosan was added, and the homogenization was carried out as in the case of BNCSFa.

The freeze-drying process was carried out as described in Section 2.2. The samples were analyzed in their initial state, excluding the analyses where it was mentioned that the samples were freeze-dried.

### 2.4. Physical-Chemical Characterization of SeBNCSFa

#### 2.4.1. Transmission Electron Microscopy (TEM)-Energy Dispersive X-ray (EDX) Analysis

TEM (TECNAI F20 G2 TWIN Cryo-TEM (FEI) transmission electron microscope, Houston, TX, USA) was used for the analysis of CSFa(−), CSFa(+), and SeBNCSFa. The freeze-dried samples were resuspended in ddH_2_O (125 mg/L). After the addition of a 10 µL sample to a carbon type-B, 200 mesh copper grid (Ted Pella, Redding, CA, USA), the grid was dried at room temperature. The images were acquired using an electron acceleration voltage of 200 kV. The elemental analysis was assessed using the EDX detector (X-MaxN 80T—Oxford Instruments, Abingdon, Oxfordshire, UK).

#### 2.4.2. Scanning Electron Microscopy (SEM) Analysis

SEM micrographs of freeze-dried CSFa(−), CSFa(+), BNC, and SeBNCSFa were acquired using a TM4000Plus II tabletop electron microscope (Hitachi, Tokyo, Japan). The following parameters were used: 15 kV electron acceleration voltage, 1000× magnification, backscattered electrons (BSE), and standard vacuum (M) mode. The adjustments were made according to the manufacturer’s recommendations.

#### 2.4.3. Fourier Transform Infrared Spectroscopy (FTIR) Analysis

The FTIR spectra of CS, freeze-dried CSFa(+), and SeBNCSFa were recorded using an IRTracer-100 spectrometer (Shimadzu, Kyoto, Japan). The analyses were assessed in attenuated total reflectance (ATR) mode as mean of 45 scans with a resolution of 4 cm^−1^ in the mid-IR spectral range of 4000–400 cm^−1^.

#### 2.4.4. X-ray Diffraction (XRD) Analysis

XRD analysis of CS, Fa, freeze-dried CSFa(+), and SeBNCSFa was performed using a Rigaku-SmartLab diffractometer (Rigaku, Tokyo, Japan). The wide-angle X-ray scattering (WAXS) was performed at 45 kV voltage, 200 mA intensity, using Cu_Kα1_ incident radiation with 1.54059 Å wavelength in the Bragg’s angle range 2θ between 2 and 90°, with 4°/min scanning speed and 0.02° resolution. The characteristic peaks and crystallinity were determined with the PDXL 2.7.2.0 software and graphically represented with the OriginPro2022b software, version 9.9.5 from OriginLab Corporation (Northampton, MA, USA).

#### 2.4.5. Thermogravimetric Analysis (TGA)

TGA analysis of CS, Fa, freeze-dried CSFa(+), and SeBNCSFa was assessed using a Q5000IR instrument (TA Instruments, New Castle, DE, USA) with the addition of about 1–4 mg of sample in the platinum pan. The samples were heated under nitrogen (99.99%) from 20 °C to 700 °C with a temperature ramp of 10 °C/min. At 700 °C, the purge gas was changed to synthetic air (99.99%) and kept isothermal for 10 min in order to assess the ash content.

#### 2.4.6. Rheology and Adhesion Analyses

The rheological behavior of individual solutions of CS, CSFa(+), and BNC and of the final hydrogel SeBNCFa was studied using an HR20 Discovery Hybrid rotational rheometer from TA Instruments (New Castle, DE, USA), and the results were processed using the TRIOS software version 5.1.1 (TA Instruments, New Castle, DE, USA). The samples were analyzed at 25 °C using a 40 mm cylindrical geometry and a 500 µm geometry gap in oscillatory sweep and flow sweep with hysteresis (up and reverse speed variation), ending in axial mode to measure the adhesion force and adhesion energy. A sample amount of around 0.7 mL, able to fill the 500 µm gap, was first subjected to an oscillation sweep in the angular frequency range of ω 0.1–100 rad/s, followed by reverse oscillation sweep from 100–0.1 rad/s. The parameters determined by oscillatory sweep were: storage modulus (G′, Pa), loss modulus (G″, Pa), complex viscosity (η*, Pa∙s), dynamic viscosity (η′, Pa∙s), and phase angle (δ, °). The analysis continued with the linear flow sweep from shear rate 1 to 100 s^−1^, followed by reverse flow sweep from 100 to 1 s^−1^ to evaluate the thixotropy. The viscosity (η, Pa∙s) and stress σ (Pa) variation with the shear rate (γ, s^−1^) were evaluated with the best fitting model. Finally, the cylindrical geometry was raised in axial mode with a constant speed of 10 µm/s for a duration of 5–8 min. in order to determine the axial force that opposes the detachment from a quartz surface and a titanium surface. The TRIOS software was used to determine the adhesion time at the maximum axial force and also to evaluate the adhesion force and adhesion distance by interpolation. The exponential function applied to the detachment curve was used to evaluate the speed of detachment as the “c” parameter of the exponential function as in a previous study [18].

#### 2.4.7. Determination of Antioxidant Activity

Radical scavenging activity by 2,2-diphenyl-1-picrylhydrazyl (DPPH) assay [19,49] was performed by mixing 100 μL of 10 mg/mL sample (*w*/*v*)/standard with 100 μL of 0.3 mM DPPH reagent previously prepared in absolute ethanol. After incubation of the mixture at room temperature for 30 min in the dark, it was centrifuged at 6000 rcf, and the absorbance was measured at 517 nm using a microplate reader. The calibration curve was prepared starting from a stock solution of 150 µM Trolox in 70% ethanol. The results were expressed as µM Trolox equivalent (TE)/g of sample.

For the ferric-ion reducing antioxidant power (FRAP) assay [19,50], three solutions were prepared: 0.3 M sodium acetate buffer (pH = 3.6), 10 mM TPTZ in 40 mM HCl, and 20 mM FeCl_3_ in double-distilled water. These three solutions were combined in order to obtain the FRAP reagent, using a ratio of 10:1:1. The FRAP reagent was maintained at 37 °C up to the time of analysis. Over 285 µL of FRAP reagent, 15 µL of 10 mg/mL (*w*/*v*) sample/standard was added. After 30 min of incubation at 37 °C in the dark, the mixtures were centrifuged at 6000 rcf, and the absorbance was measured at 593 nm using a microplate reader. The calibration curve range was prepared starting from a stock solution of 450 µM Trolox in 70% ethanol. The results were expressed as µM Trolox equivalent (TE)/g of sample.

### 2.5. Evaluation of the Biological Activity

#### 2.5.1. Cytocompatibility Analysis of Hydrogel Formulations

Human gingival fibroblasts were maintained at 37 °C under a 5% CO_2_ atmosphere in DMEM supplemented with 10% FBS. The cells were seeded in 48-well plates at a density of 1 × 10^4^ cells/cm^2^ and incubated for 24 h. Afterwards, the cells were treated with several final concentrations of BNCSFa (10, 25, 50, 100, and 500 µg/mL, i.e., 10BNCSFa, 25BNCSFa, 50BNCSFa, 100BNCSFa, and 500BNCSFa, respectively) and BNCSFa at the same concentrations enriched with 2.5 µg/mL of SeNPsK (Se10BNCSFa, Se25BNCSFa, Se50BNCSFa, Se100BNCSFa, and Se500BNCSFa, respectively). At 24 h post-treatment, cell viability was assessed by combining the Cell Counting Kit-8 (CCK-8) assay, which quantifies the number of metabolically active viable cells, with the LIVE/DEAD staining technique, which differentiates between dead and living cells, according to [19,46]. The fluorescence microscopy images obtained after the treatment with the LIVE/DEAD Viability/Cytotoxicity Kit were acquired with the CelenaX Imaging System (Logos Biosystems, Annandale, VA, USA), equipped with CelenaX Explorer software version 1.0.5 (4× objective). The samples were accompanied by a negative cytotoxicity control (C−, untreated cells) and a positive cytotoxicity control (C+, cells treated with 7.5% dimethyl sulfoxide (DMSO)). The cytoskeleton was highlighted by labeling the actin filaments with Alexa Fluor 488-coupled phalloidin, and the nuclei were marked with DAPI [19,46]. The fluorescence microscopy images were acquired with the CelenaX Explorer (20× objective).

#### 2.5.2. Investigation of In Vitro Antioxidant Activity

The in vitro antioxidant activity was investigated 24 h after the cell treatment with the highest dose of hydrogel, which exhibited the maximum potential for increasing the number of metabolically active viable cells in the presence of a reactive oxygen species (ROS) inducer (37 µM H_2_O_2_). The same initial cell density was used as in the aforementioned tests (Section 2.5.1). In parallel, a negative control (C−, untreated cells) and a positive control (C+, cells treated with 37 µM H_2_O_2_) were used. The ROS production was quantified after cell labeling with a 2′,7′-dichlorodihydrofluorescein diacetate (H_2_DCFDA) solution [19] and reading the fluorescence intensity of the supernatant obtained after cell lysis [51]. The quantitative analysis was correlated with the fluorescence microscopy images acquired with CelenaX Explorer (4× objective) after cell labeling.

#### 2.5.3. Assessment of the Pro-Inflammatory Mediators

For the assessment of the pro-inflammatory mediators released in the culture media, the HGF-1 cells were seeded in 48-well plates at a density of 1 × 10^4^ cells/cm^2^ and incubated for 24 h (as in Section 2.5.1 and Section 2.5.2). Afterwards, treatments with the highest dose of hydrogel that exhibited the maximum potential for increasing the number of metabolically active viable cells, as well as with 2.5 µg/mL SeNPsK (the same dose with which BNCSFa is enriched), were applied in the presence of 1 µg/mL lipopolysaccharides (LPS) from *P. gingivalis*/*E. coli*. The samples were accompanied by a negative control (C−, untreated cells) and a positive control (C+, cells treated with 1 µg/mL LPS from *P. gingivalis*/*E. coli*). At 24 h post-treatment, the CCK-8 assay was performed. The pro-inflammatory cytokine level released in the culture media was quantified at 24 h post-treatment using sandwich enzyme-linked immunosorbent assays (ELISA) kits, according to the manufacturer’s instructions. The following cytokines were investigated: interleukin-6 (IL-6), tumor necrosis factor (TNF-α), and interleukin-1β (IL-1β). The absorbance was measured at 550 nm using a microplate reader. After interpolation of the optical densities (OD) of the samples using the corresponding standard curve, the cytokine content expressed in pg/mL was normalized to the OD obtained from the CCK-8 assay. The nitric oxide (NO) production was investigated at 24 h post-treatment by measuring the nitrite content in the culture media using the Griess reagent system, following the manufacturer’s instructions. The nitrite content (µM) was normalized to the OD obtained from the CCK-8 assay.

#### 2.5.4. Investigation of Prebiotic Activity

*L. reuteri* and L. *salivarius* were grown in MRS broth at 30 °C in Oxoid™ AnaeroJar™ 2.5 L (Thermo Scientific™, Waltham, MA, USA) for 72 h. Afterwards, the probiotic bacteria were subcultured on MRS agar, subjecting the plates to the same conditions. Subsequently, 180 µL of 50BNCSFa and Se50BNCFa suspensions solubilized in MRS broth were added to 96-well plates, after which 20 µL of the probiotic bacteria suspensions prepared in sterile saline solution (0.8% NaCl) and adjusted to 0.5 McFarland [1.5 × 10^8^ colony forming units (CFU)/mL] using a McFarland DEN-1B densitometer (Grant Bio, Cambridge, UK) was added. The absorbance was measured at 600 nm using a microplate reader. The samples were followed by a negative control (C−, 180 µL MRS broth, 20 µL sterile saline buffer) and a positive control (C+, 180 µL probiotic bacteria suspension, 20 µL sterile saline buffer). The experiment was performed in triplicate [52].

#### 2.5.5. Determination of Antimicrobial Activity

The semiquantitative screening of the antimicrobial activity of the selected hydrogel formulation was performed using the diffusimetric method [18,19]. The microbial suspensions were prepared in sterile saline solution (0.8% NaCl) from 18–24 h cultures grown on non-selective solid medium—MHA for *S. aureus*, *B. cereus*, *P. aeruginosa*, and *E. coli* and SDA for *C. albicans*. The suspensions were adjusted to 0.5 McFarland. The microbial strains were seeded on the surface of the solid agar medium, after which 25 µL of hydrogel formulation was spotted on the surface and in the middle of the plates. After incubation for 24 ± 2 h at 37 °C, the antimicrobial activity was determined by measuring the diameter of the clear zone of three biological replicates (average of 4 measurements per replicate) using ImageJ 1.53k software [53]. The results were reported as the average diameter of the three biological replicates ± standard error. In order to determine the bacteriostatic or bactericidal effect of SeBNCSFa, a qualitative screening was performed by scraping the clear zone obtained by the diffusimetric method using a cotton swab that was inoculated on another plate previously poured with MHA/SDA medium. The images of the plates were acquired after a 24 h incubation step at 37 °C. Absence and presence of growth indicated bactericidal/fungicide and bacteriostatic/fungistatic effects, respectively.

Quantitative screening of the antimicrobial activity was assessed at different microbial cell densities [18,19] in order to see if there is a potential to prevent microbial growth at the highest dose that exhibited the maximum potential for increasing the number of metabolically active viable cells. Therefore, suspensions of hydrogel formulations, as well as 2.5 µg/mL SeNPsK (the same dose with which BNCSFa is enriched), were prepared in MHB for *S. aureus*, *B. cereus*, and *P. aeruginosa* and in SDB for *C. albicans*. Afterwards, 180 µL of each suspension was mixed in 96-well plates with 20 µL of microbial suspensions at different cell densities (1.5 × 10^8^, 1.5 × 10^7^, 1.5 × 10^6^, 1.5 × 10^5^, 1.5 × 10^4^, 1.5 × 10^3^, 1.5 × 10^2^, 1.5 × 10^1^), which were previously prepared in 0.8% NaCl. The OD values were recorded at 600 nm using a microplate reader. Each microbial cell density was followed by a microbial culture control (C+, positive control). A sterility control (C−, negative control) was also provided.

### 2.6. Statistical Analysis and Graph Generation

Statistical analysis was performed with IBM SPSS 26 software version 26.0.0.0 (one-way ANOVA). The graphs were created with OriginPro software version 9.9.5 from OriginLab Corporation (Northampton, MA, USA), except the TGA graphs that were prepared with the equipment software, TA Universal Analysis 2000, version 4.5A, and the rheology graphs that were prepared with the equipment software, TRIOS version 5.1.1.

## 3. Results

### 3.1. Physical-Chemical Characterization of CSFa and SeBNCSFa

The intrinsic viscosity of the chitosan used in this study was 4.7136 dL/g due to the dependence of the reduced viscosity on the chitosan concentration (Appendix A). The molecular weight of the chitosan was calculated at 101.26 kDa. The deacetylation degree of initial chitosan was determined to be 86 ± 2% from ^1^H NMR (Appendix A). CSFa(−) and CSFa(+) had a deacetylation degree of 86.4 ± 2% and 86.7 ± 2%, respectively (Appendix A), which indicated that the deacetylation degree remained constant throughout the grafting process. The ferulic acid grafted onto chitosan determined by the Folin–Ciocalteu method was 16.70 ± 0.39 mg ferulic acid/g chitosan. The color variation of chitosan solutions following the ninhydrin assay is depicted in Appendix A. The maximum absorption peak was recorded at 580 nm (Appendix A). Following the quantification of free amino groups, the D-glucosamine content of initial chitosan was 228.77 ± 5.84 mg/g chitosan. A significant decrease was observed for CSFa(−), reaching 151.52 ± 0.44 mg/g, followed by a subsequent significant decrease in the CSFa(+) sample down to 128.83 ± 1.25 mg/g, the difference recorded between CSFa(−) and CSFa(+) samples being 22.69 ± 0.80 mg D-glucosamine/g chitosan (Appendix A). This is equivalent to 24.66 ± 0.86 mg Fa/g chitosan, which was considered as Fa grafted to the amino groups of chitosan. The grafting of ferulic acid to chitosan was also confirmed by UV–Vis analysis (Appendix A). Based on the average value of 20.68 ± 0.63 mg Fa/g chitosan between the Folin–Ciocalteu and ninhydrin assays, a molar ratio of 10.78 ± 0.45 grafted Fa:chitosan was obtained.

The grafting degree (percent of grafted chitosan monomer units of the total (acetylated and non-acetylated) chitosan monomer units) was approximately 1.780 ± 0.07% (17.8% efficiency). Reported only to the deacetylated monomer units, the percent is 2.07 ± 0.07%, considering grafting only to amino groups.

Based on Pade Laplace (PD) as well as the SBL simulation method, the DLS analysis showed that CS had a large hydrodynamic radius (an average of 7503 ± 2453 nm diameter) in 2% acetic acid, as expected for long, disordered chains (Appendix A). CSFa(−) presented particles with much smaller diameters both in 2% acetic acid (80 nm by PD method; 40 and 117 nm by SBL method) and in water (157 nm by PD method; 89 and 170 nm by SBL method), whereas the large structures observed in CS were negligible (Appendix A). CSFa(+) had slightly larger diameters than CSFa(−): 178 nm by the PD method and 102 and 167 nm by the SBL method in 2% acetic acid; 52 and 201 nm by the PD method and 79 and 99 nm by the SBL method in water (Appendix A). The polydispersity index (PDI) was 0.34, 0.30, and 0.24 for CS, CSFa(−), and, respectively, CSFa(+) in 2% acetic acid, and 0.28 and 0.36 for CSFa(−) and, respectively, CSFa(+) in water (Appendix A). The zeta potential was 30.66 ± 1.12, 22.38 ± 3.54, and 25.69 ± 3.91 for CS, CSFa(−), and, respectively, CSFa(+) in 2% acetic acid, and 12.66 ± 2.95 and 16.38 ± 0.61 for CSFa(−) and, respectively, CSFa(+) in water (Appendix A).

TEM analysis indicated that grafting the chitosan with ferulic acid led to the formation of spherical nanoparticles with a diameter between 100 and 200 nm (Figure 1b). CSFa(−) did not present these structures but showed the presence of some small aggregates of approximately 20 nm (Figure 1a), as well as larger aggregates and long chitosan fibers, which were less visible in CSFa(+) (Appendix A). After the solubilization of CSFa in the BNC-SeNPsK suspension, a structural rearrangement imposed by the hydrophilic nature of BNC occurred, resulting in a mesh-like structure with numerous nanofilaments (Figure 1c). By embedding SeNPsK into the BNCSFa matrix, a selenium-enriched hydrogel was obtained, according to the EDX analysis (Figure 1d).

SEM micrographs revealed that CSFa(−) and CSFa(+) presented a lacy structure (Figure 2a,b). BNC has a disordered fibrillar structure combined with partially ordered micro-sheets (Figure 2b). In contrast, the BNCSFa matrix with embedded SeNPsK is a well-ordered fibrillar structure with parallel striations (Figure 2c).

FTIR spectroscopy evidenced the particularities of chitosan grafting with ferulic acid and the molecular interactions between the main biocompounds in the final SeBNCSFa hydrogel. In the diagnostic region 3600–3000 cm^−1^, specific for the hydrogen bonds stretching vibrations, the shifts of chitosan peaks from 3354 and 3289 cm^−1^ to lower wavenumbers, i.e., approximately 3345 and, respectively, 3260 cm^−1^ in CSFa(+), and 3343 and, respectively, 3271 cm^−1^ in the SeBNCSFa hydrogel, suggest newly formed HO..H and HN..H hydrogen bonds [54,55]. In the fingerprint region, the amide I, II, and III bands in chitosan are visible at approximately 1647, 1589, and, respectively, 1319 cm^−1^ in Figure 3a. These bands correspond to C=O, N–H, and C–N vibrations in residual acetyl groups [54,56].

The three amide bands in initial chitosan were displaced to 1630, 1553, and, respectively, 1306 cm^−1^ in the grafted chitosan, suggesting the involvement of the corresponding groups in covalent and/or hydrogen interactions, with possible ionization [57], of the acetyl and amino groups of chitosan with the hydroxyl and carboxyl groups of ferulic acid. The bands at 1420 and 1458 cm^−1^, assigned to –CH_2_ and –CH_3_ in chitosan [56,58], disappear/are shifted to a single band at 1406 cm^−1^ in the grafted chitosan. These shifts appear in the final hydrogel, SeBNCSFa, as well. The bacterial nanocellulose has a similar spectrum to that of chitosan, without the C=O, C–N, and N–H in the amide bands. The small band at approximately 1753 cm^−1^ in CSFa(+) and SeBNCSFa was absent in chitosan but present and shifted in the activated chitosan, i.e., CSFa(−), at 1750 cm^−1^. This small shift of 3 cm^−1^ suggests that a small fraction of ferulic acid could have formed ester bonds between its carboxyl group and the hydroxyl groups of chitosan [54]. In the polysaccharide region 1200–850 cm^−1^, the main absorption peak at 1024 cm^−1^ in chitosan, corresponding to C–OH bending vibrations, is slightly shifted to 1028 cm^−1^ in CSFa(+) and to 1026 cm^−1^ in SeBNCSFa. This shift suggests the participation of the hydroxyl functional groups of chitosan and/or nanocellulose in molecular interactions with the functional groups of ferulic acid. Moreover, the O–C and C–N absorption bands at 991 and 947 cm^−1^, respectively, in neat chitosan disappear in CSFa(+) and in SeBNCSFa spectra by convolution with the shifted C–OH bands around 1028 and 1026 cm^−1^, strengthening the hypothesis of multiple hydrogen bonds between the hydroxyl and carboxyl groups of ferulic acid and the hydroxyl and amino groups of nanocellulose and chitosan interconnected in a possible semi-interpenetrating biopolymeric network [58].

The X-ray diffraction patterns at wide angles (WAXS) of chitosan, ferulic acid, CSFa(+), and the final hydrogel SeBNCSFa are presented in Figure 3b. Ferulic acid is a crystalline compound and showed three main peaks at 9.02° for the plane (h,k,l) (0,1,1), 12.82° for (0,2,1), and 15.66° for (0,1,2), similarly with the diffraction pattern available in the PDXL database with the PDF Card No. 00-041-1606. Chitosan is a semi-crystalline polymer with variable origin, molecular weight, and crystallinity. The chitosan from fungal chitin with medium molecular weight used in this work was similar to the pattern in PDF Card No. 00-067-1540 and showed a medium crystallinity of 48% (Xc,%) in comparison, for example, with an 83% crystallinity for a high molecular weight chitosan obtained from crab shell chitin [18]. The two main diffraction peaks of fungal chitosan appearing around 2θ angles 11.08° and 20.02° represent the convolution of two crystalline peaks around 10.87° and 20.05° with the amorphous broad peak centered around 19.36°, as resulted from the PDXL software. The grafted chitosan CSFa(+) shows a reduced crystallinity degree of 37% and three peaks around 9.12°, 13.02°, and 19.84°. The first two small peaks could indicate a new arrangement of the first chitosan peak at 11.08° upon grafting with ferulic acid. The final hydrogel SeBNCSFa contains additionally bacterial nanocellulose, of which we previously showed that it had 92% crystallinity [46], therefore the crystallinity of the lyophilized hydrogel increased up to 45%. Moreover, two new diffraction peaks appeared, a small one at 14.24° corresponding to the cellulose Iα peak at 14.26° (PDF Card No. 00-056-1719) and a second intense peak at 21.94° corresponding to the main cellulose Iα peak at 21.80°. A semi-interpenetrating polymer network hydrogel based on nanocellulose and chitosan grafted with glutaraldehyde showed an increased amorphous character combined with small peaks from nanocellulose in a previous study [58]. Our XRD analysis suggests a structural rearrangement imposed by the hydrophilic bacterial nanocellulose as a semi-interpenetrating biopolymeric network in the final hydrogel.

The TGA analysis presented in Figure 4 was performed on the final hydrogel SeBNCSFa and grafted chitosan CSFa(+) in comparison with the initial compounds. Fungal chitosan showed a small weight loss of interstitial water of around 9.41%, followed by the main weight loss of 48.25% between 100 and 400 °C, a further weight loss of 11.67% between 400 and 700 °C, and ending with a 30.67% residue in N_2_, respectively, 1.12% ash (Figure 4a). The derivative thermogram (DTG) showed the fastest weight loss at 298.8 °C and represents the characteristic decomposition temperature of the used fungal chitosan. Ferulic acid presented a characteristic decomposition temperature of 244.2 °C, with a 96.45% weight loss between 150 and 300 °C (Figure 4b).

CSFa(+) showed an interstitial water content of 10.17%, followed by a second broad weight loss step of 12.74% between 100 and 250 °C that might be relevant for the ferulic acid grafted on chitosan, considering the characteristic decomposition temperature of Fa at 244.2 °C. The third weight loss of CSFa(+) represents 43.88% and corresponds to a maximum decomposition temperature of 295.6 °C. This temperature is lower than the one for neat chitosan, which is 298.8 °C. The continuous DTG shape between the second (100 and 250 °C) and the third (250 and 400 °C) thermal regions, without a strong differentiation, is in accordance with a covalent linkage of ferulic acid to chitosan and suggests a semi-cooperative decomposition with sequential steps corresponding to different regions of the molecule: the grafted ferulic acid decomposing first and gradually continuing with the grafted segments of chitosan and further with the non-grafted chitosan regions. If we subtract from the weight loss of neat chitosan in the temperature range 250–400 °C (48.25%) the weight loss of CSFa(+) in the same range (43.88%), we obtain a 4.37% difference that might be related to the grafted segments of chitosan translated into the lower thermo-region by grafting with Fa (Figure 4c).

The lyophilized hydrogel SeBNCSFa showed an 8.89% interstitial water content and four thermal populations with characteristic temperatures around 159.9 °C, 223.8 °C, 273.4 °C, and 600.1 °C. The first three decomposition temperatures are related to bacterial nanocellulose, grafted ferulic acid, and, respectively, chitosan. The TGA and DTG curves, correlated with the XRD and FTIR results, suggest a semi-interpenetrating biopolymeric network. A very low residue (0.69%) in nitrogen was observed for SeBNCSFa (Figure 4d).

The rheology behavior was studied in oscillatory mode, flow sweep, and axial detachment for the final hydrogel SeBNCSFa, as well as for the individual suspensions, respectively, 7% CS, 7% CSFa(+), and 0.4% BNC. G′ was lower than G″ of the chitosan samples (Appendix A), which indicates a viscous (liquid-like) character induced by CS. The CS suspension in Appendix A has an almost linear behavior without hysteresis, with the loss and storage modulus increasing from G′ = 1.7 Pa and G″ = 4.8 Pa at ω = 0.1 rad/s to G′ = 324.8 Pa and G″ = 398.4 Pa at ω = 100 rad/s. The complex viscosity η* of CS suspension decreased from 51.32 Pa∙s to 5.14 Pa∙s, the dynamic viscosity decreased from 48.3 Pa∙s to 3.98 Pa∙s, and the phase angle δ decreased from 70.36° to 50.77° when going from ω = 0.1 rad/s to ω = 100 rad/s. This suggests a general pseudoplastic (shear-thinning) behavior of CS suspension.

CSFa(+) showed a turbulent noisy, behavior even in triplicate analysis, with the trend curves representing the mean behavior (Appendix A). The initial viscosity of CSFa(+) at 0.1 rad/s was 100 times lower than the initial viscosity of the CS suspension, with values at approx. 0.5 Pa∙s for both complex viscosity and dynamic viscosity. The initial behavior is liquid like, with the loss modulus larger than the storage modulus, up to 26.3 rad/s, when the crossover point occurs at crossover modulus G′ = G″ = 0.1 Pa. This modification of rheological behavior clearly suggests that chitosan suffers significant molecular changes by grafting with ferulic acid.

The storage modulus G′ was larger than the loss modulus G″ of the BNC suspension (Appendix A). This suggests an elastic (solid-like) character induced by BNC. A similar observation is valid for SeBNCSFa (Figure 5a, shown as well in Appendix A for easier comparison). The BNC suspension showed viscoelastic behavior induced by the nanofibrillar structure, with the storage modulus G′ = 10.2 Pa and loss modulus G″ = 3.2 Pa at ω = 0.1 rad/s, respectively, G′ = 26.4 Pa and G″ = 1.9 Pa at 100 rad/s. At 39.8 rad/s, the loss modulus starts to decrease and the storage modulus to increase, probably due to fibril entanglement. SeBNCSFa has a viscoelastic behavior induced by the nanofibrilar structure of BNC and an intermediary viscosity between BNC and CS suspensions. The loss modulus of SeBNCSFa has a similar decrease to that of the BNC suspension at angular frequencies higher than 39.8 rad/s.

In flow sweep mode presented in Appendix A, the CS suspension showed a Bingham pseudoplastic (shear thinning) and reopectic (anti-thixotropic) behavior (Appendix A). This means it is a fluid with yield stress and flows with viscosity decreasing when shear rate increases, whereas with time the viscosity slightly increases (reverse curve R). The viscosity curve was best fitted by the Carreau–Yasuda model, and the stress curve was best fitted by the Herschel–Bulkley model. CSFa(+) had a similar behavior, fitted by the Carreau–Yasuda model for viscosity and the Casson model for stress, but with lower viscosity and thixotropy than CS (Appendix A). The BNC suspension had the highest thixotropy and the highest yield stress among the samples (Appendix A). The final SeBNCSFa hydrogel is a Bingham pseudoplastic fluid with a yield stress of around 0.45 Pa, an intermediary viscosity between CS and BNC suspensions, and a thixotropic behavior induced by the BNC nanofibrils (Figure 5b, shown in Appendix A as well).

The adhesion properties were determined in axial mode on a standard quartz surface and on a titanium surface for the SeBNCFa hydrogel and its components. The adhesion force of the SeBNCSFa hydrogel was 0.222 N on quartz (Figure 5c and Appendix A) and 0.105 N on Ti (Figure 5d and Appendix A), the maximum adhesion time was 3 s on quartz and 8 s on Ti, and the speed of detachment was 39 µm/s on quartz and 59 µm/s on Ti. These values generally suggest good adhesion on both types of surfaces, stronger on quartz than on titanium. A longer adhesion time on Ti than on quartz might suggest more adhesion points on Ti than on quartz. If we compare the adhesion properties of the individual components in Appendix A, we can observe that the adhesive properties are mainly induced by chitosan. The grafted chitosan CSFa(+) loses a significant part of the adhesive properties of plain chitosan CS, with the adhesive force for CSFa(+) being 0.260 N on quartz (Appendix A) and 0.048 N on Ti (Appendix A), compared with the adhesive force for CS, 1.599 N on quartz (Appendix A) and 0.648 N on Ti (Appendix A). All suspensions have longer adhesion times on Ti than on quartz, respectively, 12 s on Ti and 5 s on quartz for CS, 5 s on Ti and 2 s on quartz for CSFa(+), 8 s on Ti (Appendix A), and 3 s on quartz (Appendix A) for BNC. The adhesion values of the final hydrogel SeBNCSFa are closer to the values of BNC suspension, suggesting that the outer layer of the hydrogel consists mainly of BNC, whereas the grafted chitosan micelles are embedded inside the hydrogel.

The antioxidant activity (AOA) of the SeBNCSFa hydrogel was significantly higher than that of BNCSFa for both methods employed. By DPPH assay, SeBNCSFa showed an AOA of 7.049 ± 0.339 µmol TE/g compared to 5.914 ± 0.172 µmol TE/g for BNCSFa. By the FRAP method, SeBNCSFa had an AOA of 8.245 ± 0.332 µmol TE/g, compared to 7.129 ± 0.289 µmol TE/g for BNCSFa (Figure 6).

### 3.2. Cytocompatibility Behaviour, Antioxidant and Anti-Inflammatory Potentials

The SeNPsK-free hydrogel, i.e., the BNCSFa matrix, proved to be cytocompatible for all the concentrations tested (Figure 7a), with no significant changes in cell viability compared to the cytotoxicity negative control (C−). With the addition of 2.5 µg/mL SeNPsK, there is a significant increase in the number of metabolically active viable cells, with the maximum potential being reached in the cases of Se25BNCSFa (108.6 ± 1.58% of C− cell viability) and Se50BNCSFa (107.7 ± 2.16% of C− cell viability). By increasing the Se-enriched hydrogel concentration, the cell viability starts to decrease slowly; however, these values are still higher when compared to the negative cytotoxicity control (Figure 7b). The results obtained by the CCK-8 assay can be correlated with the fluorescence microscopy images acquired after cell labeling with calcein acetoxymethyl ester and ethidium homodimer-1 in order to differentiate between live (green fluorescence) and dead (red fluorescence) cells (Figure 7c–f).

The concentration of 50 µg/mL was the highest dose of BNCSFa that induced the most statistically significant increase in the number of metabolically active viable cells in the presence of 2.5 µg/mL SeNPsK. The difference between the 25 and 50 µg/mL concentrations was not significant. Therefore, further investigations, such as analysis of cell morphology, anti-inflammatory, and antioxidant activities, as well as the analysis of the microbial growth reduction potential at different microbial cell densities, were performed for Se50BNCSFa.

There were no morphological changes following the cell treatment with 50BNCSFa (Figure 7h) or Se50BNCSFa (Figure 7i) compared to untreated cells (Figure 7g), as indicated by labeling the actin filaments and the nuclei. The actin cytoskeleton was well organized in a fibrillar structure with stress fibers distributed along the long axis of the cell body, and the fibroblasts maintained their characteristic phenotype.

Following the labeling and quantification of total intracellular ROS, a significant decrease in ROS level was observed at 24 h after the treatment with 50BNCSFa, i.e., 74.69 ± 1.78% of C−), and a 2-fold decrease compared to the positive control (C+, ROS-inducing agent). The treatment with Se50BNCSFa induced the highest decrease in the amount of ROS in the presence of the ROS-inducing agent, i.e., 41.47 ± 0.64% of C−, and a 4-fold decrease compared to C+ (Figure 8a). The quantitative results can be correlated with the fluorescence microscopy images acquired after cell labeling with H_2_DCFDA (Figure 8b–e).

Cell viability following SeNPsK, 50BNCSFa, and Se50BNCSFa treatments was studied in the presence of lipopolysaccharides (LPS) from *P. gingivalis* and *E. coli*. An increase in the number of metabolically active viable cells was observed following the treatment with 2.5 µg/mL SeNPsK in the presence of LPS from *P. gingivalis* (105.7 ± 0.78% of C−). The cell treatments with 50BNCSFa, Se50BNCSFa, and the positive control for inflammation (1 µg/mL LPS) did not induce changes in cell viability compared to the negative control for inflammation. A statistically significant decrease in cell viability was observed following the treatment with 1 µg/mL LPS from *E. coli* (84.4 ± 1.16% of C−). The 50BNCSFa matrix treatment led to a cell viability of 91.49 ± 1.04% of C−, which is still lower than the negative control but higher than the positive control for inflammation. For the cell treatments with 2.5 µg/mL SeNPsK and Se50BNCSFa, no changes in cell viability were observed in comparison with C− (Figure 9a).

The level of pro-inflammatory mediators released in the culture media was also assessed at 24 h post-treatment. An approximately 6-fold increase in IL-6 was observed for the inflammation-positive control C+ (1296 ± 4.74 pg/mL) induced by LPS of *P. gingivalis* when compared to the inflammation-negative control C− (211.6 ± 11.74 pg/mL). The treatments with 2.5 µg/mL SeNPsK and Se50BNCSFa led to a significant decrease in the IL-6 concentration when compared to the positive control for inflammation, i.e., 994.5 ± 8.79 pg/mL for SeNPsK and 949.3 ± 9.67 pg/mL for Se50BNCSFa, with no statistical differences between the two treatments. The highest decrease in the IL-6 level was observed following the treatment with the 50BNCSFa matrix, i.e., 827.3 ± 2.49 pg/mL (Figure 9b).

The tumor necrosis factor (TNF-α) concentration was 17.6 ± 2.44 pg/mL for the negative control (C−). After the treatment with 1 µg/mL LPS from *P. gingivalis*, an increase in TNF-α concentration of about 2.5 times (42.91 ± 0.83 pg/mL) compared to C− was observed. A significant decrease in TNF-α level to 30.53 ± 0.77 pg/mL was recorded for the 2.5 µg/mL SeNPsK treatment. In the case of 50BNCSFa and Se50BNCSFa treatments, the TNF-α concentrations were slightly lower compared to the inflammation positive control, but the differences were not statistically significant.

Following the treatment with 1 µg/mL LPS from *E. coli*, the concentration of TNF-α (39.2 ± 1.93 pg/mL) was similar to the concentration recorded after the treatment with 1 µg/mL LPS from *P. gingivalis*. The treatment with 50BNCSFa did not lead to a significant decrease in the level of TNF-α (36.17 ± 1.78 pg/mL) compared to the positive control for inflammation. The treatment with 2.5 µg/mL SeNPsK led to a significant decrease in the TNF-α concentration (28.78 ± 2.42 pg/mL), and Se50BNCSFa decreased further the TNF-α concentration, i.e., 23.13 ± 1.61 pg/mL, which was statistically similar to the concentration recorded in the case of C− (Figure 9c).

The concentration of interleukin-1β (IL-1β) in the media of untreated cells (C−) was 2.36 ± 0.25 pg/mL. An approximately 2-fold increase compared to C− was observed for the treatment with 1 µg/mL LPS from *P. gingivalis*. The treatment with 2.5 µg/mL SeNPsK led to a decrease in IL-1β concentration to 1.38 ± 0.12 pg/mL, slightly lower than C−. Following the treatment with 50BNCSFa, the level of IL-1β decreased significantly compared to the positive control for inflammation, reaching 0.44 ± 0.12 pg/mL, which was significantly lower than the concentration of IL-1β recorded in the case of C−. The treatment with Se50BNCSFa induced the lowest decrease in the concentration of IL-1β (1.96 ± 0.12 pg/mL), which was statistically similar to C−. Following the treatment with LPS from *E. coli*, the IL-1β level increased about 2.5 times in comparison with the negative control for inflammation. After the treatments with 2.5 µg/mL SeNPsK, 50BNCSFa, or Se50BNCSFa, the presence of IL-1β was not detected (Figure 9d).

NO production was investigated by determining the nitrite content in the culture media. In the case of the negative control (C−), a level of 1.16 ± 0.06 µM was recorded. The stimulation of human gingival fibroblasts with 1 µg/mL LPS from *P. gingivalis* produced a statistically non-significant increase in the NO level. Similar values were obtained following the treatments with 2.5 µg/mL SeNPsK, 50BNCSFa, and Se50BNCSFa in the presence of 1 µg/mL LPS from *P. gingivalis* (Figure 9e). In contrast, treatment with 1 µg/mL LPS from *E. coli* induced a statistically significant increase in the NO level when compared to C− up to 1.96 ± 0.16 µM. The treatment with Se50BNCSFa led to a significant decrease in the NO level compared to 1 µg/mL LPS from *E. coli*, which reached 1.43 ± 0.12 µM. The treatments with 2.5 µg/mL SeNPsK or 50BNCSFa produced marginally statistically significant decreases in the NO level (1.50 ± 0.06 µM for SeNPsK and 1.69 ± 0.11 µM for 50BNCSFa) in comparison with the positive control for *E. coli* LPS-induced inflammation (1.96 ± 0.16 µM). It is worth mentioning that the NO level induced by Se50BNCSFa is slightly lower than each of those induced by SeNPsK and 50BNCSFa, indicating a possible synergic effect between SeNPsK and 50BNCSFa.

### 3.3. Prebiotic Activity of Hydrogel Formulations

In the case of 50BNCSFa, there was an increase in *L. reuteri* growth at 24 h post-treatment up to 157.00 ± 2.14% of C+. A decrease in the probiotic bacteria growth was observed following the treatment with Se50BNCSFa (116.00 ± 0.54% of C+), but this value is still higher when compared to the C+ (*L. reuteri* in MRSB without any supplementation). The same pattern was recorded at 48 and 72 h post-treatment. However, the difference between 50BNCSFa and Se50BNCSFa is smaller at 48 h and 72 h than at 24 h because of a decrease in growth in the presence of 50BNCSFa. The treatment with 50BNCSFa/Se50BNCSFa has a constant effect on probiotic growth from 48 h to 72 h (Figure 10a). Between 48 h and 72 h post-treatment, *L. reuteri* entered the stationary phase of bacterial growth in both treated and C+ cultures.

In the case of *L. salivarius*, there was an increase in probiotic growth up to 145.80 ± 1.89% of C+ following the treatment with 50BNCSFa with a subsequent significant increase up to 178.40 ± 3.12% of C+ in the case of Se50BNCSFa at 24 h post-treatment. This indicates an additional prebiotic effect of SeNPsK on *L. salivarius*. After 48 and 72 h post-treatment, there are no significant differences between 50BNCSFa, Se50BNCSFa, and C+ (Figure 10b). *L. salivarius* entered the stationary phase of bacterial growth in both treated and C+ cultures between 48 h and 72 h post-treatment, similarly to *L. reuteri*.

### 3.4. Antimicrobial Potential of Hydrogel Formulations

For the quantitative screening of antimicrobial activity, the density of microbial cells was varied between 1.5 × 10^7^ and 1.5 × 10^1^. The concentration of 1.5 × 10^0^ microbial cells represented the negative control. The antimicrobial potential of the hydrogel formulation Se50BNCSFa was investigated at 12 h and 24 h post-treatment and compared with the effects of SeNPsK and 50BNCSFa treatments (Figure 11 and Figure 12).

At 12 h post-treatment, *S. aureus* grew at microbial densities between 1.5 × 10^7^ and 1.5 × 10^4^. At 1.5 × 10^7^ microbial cells/mL, the treatment with 2.5 µg/mL SeNPsK and Se50BNCSFa induced an inhibition of the microbial growth of about 20%. The antimicrobial potential of the 50BNCSFa formulation was slightly higher (24.64 ± 0.62% of C+). At 1.5 × 10^6^ microbial cells/mL, the same pattern as in the previous microbial density was observed. The SeNPsK treatment led to an inhibition of the microbial growth of 32.04 ± 3.13% of C+, which is similar to the effect observed in the case of the Se50BNCSFa treatment (30.73 ± 1.67% of C+). The antimicrobial effect of the 50BNCSFa treatment was significantly higher (45.84 ± 2.11% of C+). At 1.5 × 10^5^ and 1.5 × 10^4^ microbial cells/mL, the antimicrobial potential of all the treatments is similar, with the microbial growth inhibition being approximately 60% (Figure 11a). At 24 h post-treatment, *S. aureus* grew at microbial densities between 1.5 × 10^7^ and 1.5 × 10^3^. At 1.5 × 10^7^ microbial cells/mL, the most effective treatments were 50BNCSFa and Se50BNCSFa, with the microbial growth inhibition reaching approximately 11% of C+. The treatment with 2.5 µg/mL SeNPsK induced a lower microbial growth inhibition (6.61 ± 0.40% of C+) than the other two treatments. A similar pattern was observed for the microbial density of 1.5 × 10^6^. The treatments with 50BNCSFa and Se50BNCSFa induced an inhibition of the microbial growth of approximately 60% of C+, and the antimicrobial potential of SeNPsK was slightly lower (approximately 55% of C+). At lower microbial densities, the antimicrobial potential of all the treatments was similar. The microbial growth inhibition was saturated at approximately 70% of C+ between 1.5 × 10^5^ and 1.5 × 10^3^ microbial cells/mL (Figure 11b).

*B. cereus* grew at microbial inoculations between 1.5 × 10^7^ and 1.5 × 10^3^ microbial cells/mL. The treatments had lower antimicrobial activity against *B. cereus* than against *S. aureus*, and their behavior was different, i.e., 50BNCSFa had much lower antimicrobial activity than SeNPsK and Se50BNCSFa. Due to SeNPsK, the antimicrobial potential of Se50BNCSFa was 2-fold higher in comparison with the 50BNCSFa matrix at all the microbial densities tested. Se50BNCSFa had a higher inhibition than SeNPsK as well, due to the contribution of the antimicrobial effect of 50BNCSFa. The microbial growth inhibition following the Se50BNCSFa treatment was 9.10 ± 0.03% of C+ at 1.5 × 10^7^ microbial cells/mL, and it reached 19.38 ± 0.17% of C+ at 1.5 × 10^3^ microbial cells/mL 12 h post-treatment (Figure 11c). The pattern was similar 24 h post-treatment, but whereas the antimicrobial potential of SeNPsK slightly increased from 12 h, that of 50BNCSFa and Se50BNCSFa decreased compared to 12 h. Therefore, the microbial growth inhibition following SeNPsK and Se50BNCSFa became similar, with no significant differences in terms of the antimicrobial potential. The treatment with 2.5 µg/mL SeNPsK or 50 µg/mL Se50BNCSFa induced an inhibition of the microbial growth of approximately 7% at 1.5 × 10^7^ microbial cells/mL, which was significantly higher in comparison with the 50BNCSFa treatment (1.03 ± 0.03% of C+). The microbial growth inhibition induced by the SeNPsK and Se50BNCSFa treatments reached approximately 13% of C+ at 1.5 × 10^3^ microbial cells/mL, whereas it was 4.82 ± 0.40% of C+ for the 50BNCSFa treatment (Figure 11d).

*P. aeruginosa* grew at microbial inoculations between 1.5 × 10^7^ and 1.5 × 10^2^. At 12 h post-treatment, the Se50BNCSFa hydrogel was not very effective at microbial cell densities between 1.5 × 10^7^ and 1.5 × 10^5^ microbial cells/mL. The microbial growth inhibition ranged from 4.38 ± 0.59% to 7.52 ± 0.35% of C+, with higher inhibition at lower microbial densities. At 1.5 × 10^4^ microbial cells/mL, the antimicrobial potential of Se50BNCSFa further increased to 15.85 ± 0.53%, with subsequent increases at 1.5 × 10^3^ (20.36 ± 1.76% of C+) and 1.5 × 10^2^ microbial cells/mL (28.88 ± 0.11% of C+). There were no major differences between the three treatments, i.e., SeNPsK, 50BNCSFa, and Se50BNCSFa, in general (Figure 11e). No inhibition of the microbial growth at 24 h post-treatment was observed in the case of *P. aeruginosa*.

*C. albicans* grew at microbial densities between 1.5 × 10^7^ and 1.5 × 10^3^. At 12 h post-treatment, Se50BNCSFa presented the highest potential for inducing microbial growth inhibition in comparison with the SeNPsK and 50BNCSFa treatments. At the highest microbial inoculation, 1.5 × 10^7^, the antimicrobial potential of Se50BNCSFa was 9.96 ± 0.71% of C+, and it significantly increased as the inoculated microbial density decreased, reaching 75.76 ± 1.15% of C+ at 1.5 × 10^3^ microbial cells/mL (Figure 12a). At 24 h post-treatment, the microbial growth inhibition was reduced in comparison with the values recorded after 12 h. At the highest microbial densities, between 1.5 × 10^7^ and 1.5 × 10^5^, the microbial growth inhibition was below 10% following the Se50BNCSFa, SeNPsK, or 50BNCSFa treatment. Below 1.5 × 10^4^ microbial cells/mL, the antimicrobial potential of Se50BNCSFa increased up to 18.35 ± 0.53% of C+ at 1.5 × 10^3^ microbial cells/mL (Figure 12b).

The results of the semiquantitative screening of antimicrobial activity are presented in Table 1. The undiluted SeBNCSFa hydrogel inhibited the microbial growth of all the tested strains.

By assessing the bacteriostatic or bactericidal effect of SeBNCSFa, it was observed from Appendix A that SeBNCSFa presented a bactericidal effect against *S. aureus* and *C. albicans* and a bacteriostatic effect against *B. cereus*, *P. aeruginosa*, and *E. coli*.

## 4. Discussion

In the current study, a hydrogel formulation based on ferulic acid-grafted chitosan and never-dried bacterial nanocellulose enriched with SeNPs from Kombucha fermentation was developed.

The physical-chemical analyses evidenced a homogeneous hydrogel stabilized by non-covalent interactions between the bacterial cellulose nanomesh with embedded selenium nanoparticles and the chitosan covalently grafted with ferulic acid. The main redox reactions that occur during chitosan grafting in the presence of H_2_O_2_ and acidified medium with acetic acid, ascorbic acid, and ferulic acid consist of free hydroxyl and peroxide radical generation combined with acetate, ascorbate, and feruloil radical generation. These reactions lead to chitosan partial oxidation of the hydroxyl and amino groups with additional chain fragmentation at the glycosidic bonds, followed by covalent, ionic, and non-covalent recombination between oxidized chitosan and ionized molecules. The ascorbic acid and H_2_O_2_ redox pair was proven to generate ascorbate radicals without residual hydroxyl radicals. The ascorbate radicals had the main role in mediating the grafting of chitosan with caffeic acid, in comparison with the poor performance of hydroxyl radicals generated by the Fenton reaction [59]. The ascorbate radical can extract one hydrogen from a C–H bond in the chitosan chain to create a carbon-centered radical, which becomes reactive for the grafting reaction with caffeic acid [59]. In another method of chitosan grafting with gallic acid using carbodiimide as a mediator, the grafting of gallic acid was reported to occur by amide and ester linkages at the hydroxyl groups of C2, C3, and C6 carbons in chitosan [60].

The ninhydrin assay showed a decrease in reacting amino groups towards ninhrydine upon activation with H_2_O_2_ and ascorbic acid, which could be induced by the oxidation and amino radical formation. Another explanation could be related to the formation of nanostructures, as evidenced by DLS, which could partially hinder the access of ninhydrin molecules to the amino groups. The assay showed that at least part of Fa was grafted at amino groups. DLS indicated that the nanostructures are maintained upon grafting, but TEM indicated that the nanostructures of CSFa(+) are different than those of CSFa(−), i.e., defined spherical nanoparticles (NPs) compared with rather aggregates in the case of CSFa(−). Similar spherical NPs were reported for epigallocatechin-gallate-grafted chitosan [61]. It is possible that the extra hydrophobicity induced by Fa grafting causes CS oligosaccharides to fold into these symmetrical structures, with Fa probably oriented inside. The polidispersity index (PDI) decreased from CS to CSFa(−) and to CSFa(+) in acetic acid, in agreement with the formation of more compact and defined nanostructures. In water, CSFa(+) presents a higher PDI than in 2% acetic acid and is higher than the PDI of CSFa(−). Both CSFa(−) and CSFa(+) have lower zeta potential in water than in 2% acetic acid, due to a lower protonated degree and positive charge in water, indicating a higher tendency towards aggregation. The less protonated form in water coupled with the increased hydrophobicity induced by Fa, probably facilitates more inter-molecular aggregation in CSFa(+) and leads to the highest PDI (0.36) among all combinations (Appendix A). In acetic acid, the increased positive charges in NPs probably force the structures to adopt more compact and isolated NPs stabilized by internal hydrophobic interactions facilitated by the grafted ferulic acid.

FTIR spectroscopy evidenced the grafting of chitosan with ferulic acid by amide I, II, and III band shifts, which correlates with the formation of a new amide group between the -NH_2_ of chitosan and the -COOH of ferulic acid. The additional covalent ester bonds at approximately 1753 cm^−1^ suggest that a small part of ferulic acid could have interacted with the -OH of chitosan [54,60]. In a previous grafting protocol of chitosan with gallic acid, a new carbon–oxygen bond was reported at approx. 1771 cm^−1^ [62], similar to the ester bond around 1753 cm^−1^ between chitosan and ferulic acid in our work. Considering that the Folin–Ciocalteu assay gave a lower value of grafted ferulic acid than ninhydrin, it is possible that the further structural modifications within chitosan upon grafting led to an overestimation of the grafted degree of amino groups determined by ninhydrin. The formation of nanostructures could hinder the -NH_2_ groups that react with ninhydrin, as mentioned above. If and how much the overestimation is, it needs more in-depth studies. The eventual overestimation of grafting at amino could be partially compensated by the grafting to -OH groups, as suggested by FTIR. The amide I and II bands in CSFa(+) and SeBNCSFa appear shifted to lower wave numbers, agglutinated, and more intense compared with neat chitosan in Figure 3a. This suggests successful ferulic acid grafting to the deacetylated amino groups and confirms the ninhydrin assay. The chitosan band around 1458 cm^−1^ assigned to C–H coupled with N–H vibration in amines and amides [63,64] and the band around 1420 cm^−1^ assigned to –CH_2_–OH bending vibrations [18,59] are shifted to one band around 1406 cm^−1^ in grafted chitosan and SeBNCSFa hydrogel, suggesting participation in hydrogen bonds after grafting. Changes in the hydrogen bonds are suggested by the shifts in the diagnostic region, 3600–3000 cm^−1^ as well. The band at 1375 cm^−1^ is generally assigned to -CH_3_ vibration in the acetyl groups of chitosan [18,59] and remains unshifted in CSFa(+) and SeBNCSFa. The band at approximately 1319 cm^−1^ represents the “classical” amide III band, assigned to C–N vibrations combined with N–H in-plane bending, and forms, together with the other bands in the region 1400–1200 cm^−1^, the extended amide III region [65]. The amide III band appears slightly shifted to 1306 cm^−1^ in the grafted chitosan as a consequence of the structural modifications. The bands at approximately 1319 cm^−1^ and 1261 cm^−1^ are assigned to α-helix and, respectively, β-turns in proteins [66,67]. Regarding chitosan, this might suggest two particular chain arrangements, with the first arrangement being more flexible and susceptible to intermolecular interactions, similarly to the α-helix arrangement.

All in all, the results suggest as the main grafting site the amino groups, with some possible additional grafting to the -OH of chitosan, most probably at C6 as usually reported (Figure 1).

The XRD-WAXS analysis showed an increased amorphous character of the hydrogel in comparison with the semi-crystalline character of chitosan and BNC. This suggests a good miscibility at the molecular level between the biocompounds and, implicitly, a homogeneous blend. A similar increase in the amorphous character was observed for a starch–chitosan–ferulic-acid film [68], for chitosan–ferulic acid microcapsules [69], and for a chitosan suspension in acetic acid blended with poly(*N*-vinyl pyrrolidone) [70]. The increased amorphous character in these studies was similarly explained by a good miscibility between the components, with probable formation of feruloyl groups by grafting to chitosan chains [69], which hinders the individual chain arrangement in the crystalline form.

Thermogravimetric analysis allows the evaluation of the thermal stability of a (bio)material, the relative weight composition by weight loss percents in characteristic thermal regions, and the pyrolytic residue in nitrogen and, respectively, ash in air flow. The derivative curve DTG describes the particularities of the thermal events through the local speed of thermal decomposition. In Figure 4c,d, we consider that the DTG curve can be relevant for the particular (macro)molecular interactions between chitosan and ferulic acid, respectively, grafted chitosan and bacterial nanocellulose. Grafting ferulic acid to chitosan determined a decrease of 3.2 °C in the peak decomposition temperature of grafted chitosan CSFa(+) compared with neat chitosan and a partial convoluted DTG shape between the second (100–250 °C) and third (250–400 °C) thermal regions of CSFa(+). This convolution indicates a rather strong interaction between the grafted ferulic acid and chitosan, with possible destabilization of the chitosan intra- and interchain interactions. A weaker interaction might result in a clear differentiation between DTG peaks, as appeared in a previous chitosan grafted with ferulic acid using a different method, with one DTG peak around 185 °C and a second separated one at approximately 308 °C [57]. The interaction between grafted chitosan and bacterial nanocellulose in the final SeBNCFa hydrogel induces a 25.4 °C decrease in the neat chitosan decomposition temperature in Figure 4d and a more complex convoluted DTG curve between 100 and 350 °C. This suggests a strong interaction between the components of the hydrogel. The convoluted DTG curve could be explained in the following way. In our previous study related to Kombucha bacterial nanocellulose synthesis, purification, and microfluidization [20], the TGA and DTG analyses evidenced that after 25 passes at 1300 bar through the microfluidizer, two thermal populations emerged from the initial bacterial nanocellulose. The first population, with a T_max_ of approximately 250 °C, represented short nanofibrils broken by microfluidization and predominantly one-chain triclinic Iα cellulose. The second population showed a T_max_ around 345 °C, consisted of two-chain monoclinic Iβ-rich cellulose fibrils, and was comparable with the microcrystalline cellulose used in the same study. These two nanocellulose populations, one with short fibrils and the second with long fibrils, might both interact with the ferulic acid grafted on chitosan. This interaction might lead to the thermal populations with T_max_ around 160 °C and, respectively, the DTG shoulder around 330 °C not being present for CSFa(+) in Figure 4c. The other two thermal populations at approximately 224 °C and 273 °C are related to the grafted ferulic acid on chitosan, respectively, to the main chain of chitosan decomposition.

The very low residue in nitrogen of 0.69% was surprising, considering the high residues of chitosan and grafted chitosan CSFa(+), and suggests that the SeBNCSFa hydrogel is a new nano-blend of highly rearranged molecules that gradually decomposes pyrolytically with a final step at 600.1 °C and accounts for a 30.81% weight loss. The almost complete decomposition (99.31%) of SeBNCSFa in nitrogen below 630 °C might occur due to a nanoporous arrangement of the nanocellulose fibrils and chitosan grafted with ferulic acid, as suggested by the SEM micrographs as well (Figure 2), which could increase the heat propagation at the intermolecular level. The effect of this nano-arrangement in the SeBNCSFa hydrogel is possibly amplified by a pyro-catalytic effect of selenium nanoparticles due to the fast decrease in the thermal event. The decrease in thermal decomposition temperature during pyrolysis, together with a 100% thermal decomposition, was observed at 1.3% metal oxide nanoparticles added to asphaltene, the observations being correlated with a catalytic effect induced by nanoparticles [71].

The SeBNCSFa hydrogel is a soft, non-crosslinked/-reticulated hydrogel; therefore, its mechanical strength and rheological properties are lower than those of crosslinked hydrogels. The SeBNCSFa hydrogel showed a storage modulus variation between 1.8 and 8.3 Pa, whereas for crosslinked hydrogels the values are higher, for example, from 3 to 2000 Pa for a crosslinked alginate hydrogel [72] or from 17 to 25 Pa for an alginate–catechol crosslinked hydrogel [73]. Crosslinked hydrogels have additionally a high adhesion strength on various surfaces. An alginate hydrogel crosslinked with 21% dopamine showed a 7.2 kPa adhesion strength on glass determined by lap shear stress [72], and an alginate–catechol crosslinked hydrogel showed a 5.4 kPa adhesive potential corresponding to 4 J/m^2^ determined by slow retraction of the steel plate [73]. Stronger fatigue-resistant hydrogels can show resistance to forces up to 1000–10,000 N determined by peeling tests, with high interfacial toughness on various surfaces like glass, ceramic, Ti, Al, or steel [74]. In our case, the 105 mN adhesion force of the SeBNCSFa hydrogel on Ti surface is lower than the previously mentioned values for crosslinked hydrogels but considerably higher than the adhesion force of six different oral bacterial strains on Ti surface, measured by force spectroscopy and ranging between 0.2 and 2 nN [75]. The hydrogel will therefore impair the adhesion of bacterial pathogens to the Ti surface. The values are promising for the application as a soft antimicrobial and biocompatible hydrogel with good adhesion to a titanium dental implant, aiming to facilitate fast integration into the surrounding tissues.

The antioxidant properties of the hydrogel SeBNCSFa highlighted by DPPH and FRAP assays are significantly higher in comparison with the BNCSFa matrix. Despite the small quantity added to the hydrogel matrix (2.5 µg/mL), SeNPsK reduces the free radicals and the ferric ions, as Se^0^ donates electrons and gets into the oxidized forms. In our previous research [46], we investigated the antioxidant capacity of SeNPsK at three different concentrations. It ranged from 46.3 µM ± 3.1 TE at 66 µg/mL SeNPsK to 85.7 ± 0.6 µM TE at 200 µg/mL SeNPsK in the DPPH assay. For the FRAP method, the antioxidant capacity ranged from 104.4 ± 2.3 µM TE at 66 µg/mL SeNPsK to 155.28 ± 3.09 µM TE at 200 µg/mL SeNPsK. This study reveals that even at much lower concentrations, SeNPsK can make a difference in the AOA. Several research studies have been conducted in terms of the development of chitosan conjugates with ferulic acid [27,76,77]. The AOA of these conjugates depends on the hydrogen-donating ability of the conjugate product, the synthesis method—free radical-mediated grafting method [22,25,28], enzyme-catalyzed grafting method [76,78], carbodiimide based chemical coupling method [27], and the molar ratio between chitosan and ferulic acid, the grafting degree [79].

Dental implants are widely used worldwide and have a success rate of over 90%, but severe complications can occur in some cases [80]. Even though titanium is commonly used for its safety and biocompatibility, the infection-associated failure of osseointegration of Ti implants is still approximately 14% [81]. There are multiple factors that lead to this drawback of dental implants, such as the patient’s health status, a history of periodontal disease, smoking, poor oral hygiene, the type of implant material [82], etc. In such cases, the implant surface will become a support for the adhesion of pathogenic bacteria and the formation of dysbiotic biofilms. Planktonic bacteria cause acute infections, whereas bacteria in biofilms (sessile) cause chronic antibiotic-resistant infections [80]. Therefore, functional and protective coatings and interface integration products are essential for enhancing the efficacy and safety of several implants. These materials serve as barriers against microbial colonization, preventing implant-related infections [83,84] and pro-inflammatory cytokine release [85]. In a previous study, ferulic acid-grafted chitosan proved to show cytocompatibility on mesenchymal stem cells of dental pulp, and in the case of mouse fibroblasts, it demonstrated the potential to increase the number of metabolically active cells by 50% compared to the untreated cells [86]. There are also numerous studies on the cytocompatibility behavior of bacterial nanocellulose, which is additionally an excellent carrier for various biocompounds [15,87]. Moreover, nanocellulose has proven to be an efficient candidate for tissue regeneration. A hydrogel based on nanocellulose and acrylic acid led to an increase in cell attachment of more than 80% in 4 h [88]. In the current study, the BNCSFa hydrogel proved to be cytocompatible at all tested concentrations. We tested the cytocompatibility of SeNPsK with human gingival fibroblasts in our previous study at different concentrations, and the maximum potential for inducing cell proliferation was recorded in the case of 2.5 µg/mL SeNPsK treatment at 24 h post-treatment [46]. A significant increase in the number of metabolically active viable cells up to 108% of C− was observed for the hydrogel enriched with 2.5 µg/mL SeNPsK at 25 and 50 µg/mL BNCSFa (Se25BNCSFa and Se50BNCSFa treatments, respectively), indicating that the hydrogel matrix can preserve the bioactivity of SeNPsK. In another study, we investigated the in vitro antioxidant activity of a biopolymeric nanoformulation based on commercial water-soluble fungal chitosan and nanocellulose enriched with SeNPs photosynthesized by sea buckthorn leaf extract, and the biopolymeric matrix did not exhibit the potential to inhibit ROS production [19]. In the current study, the 50BNCSFa matrix reduced the amount of intracellular ROS by 2-fold compared to the positive control, due to the antioxidant potential of the grafted ferulic acid. A further decrease in ROS production by 4-fold compared to the positive control was noted in the case of Se50BNCSFa, which is similar to the in vitro antioxidant activity of 2.5 µg/mL SeNPsK investigated in our previous study [46].

The anti-inflammatory potential of Se50BNCSFa was investigated in the presence of LPS from *P. gingivalis* and *E. coli*. Bacterial LPS triggers the release of pro-inflammatory cytokines, which initiate the inflammatory process. The failure to reduce the inflammation will lead to further hyper-inflammatory responses that will result in the proliferation of dysbiotic biofilm [89]. *P. gingivalis* is a Gram-negative anaerobe, part of the red complex of periodontal pathogenic bacteria, which acts at the level of periodontal pockets [90], and it has been associated in many cases with implant failure [91]. In addition, LPS from *E. coli* was used as well, since numerous studies showed the enhanced potential of this bacterial toxin to induce an increase in the IL-6 pro-inflammatory cytokine [92]. This is associated with peri-implantitis in many cases, an inflammatory pathological condition that leads to the destruction of the soft tissues surrounding dental implants. Moreover, it was reported that IL-6, TNF-α, and IL-1β act in a synergistic manner in the osteoclastogenesis process [93], which can lead to the loss of the alveolar bone [94]. Pro-inflammatory cytokines also lead to an increased expression of inducible nitric oxide synthase (iNOS) by HGF cells, which causes an increased production of NO compared to basal conditions [95]. Since NO has a very short life time, the NO production was investigated in the current study by the assessment of the nitrite concentration, a stable end-product of the NO metabolism [96]. In the current study, LPS from *E. coli* led to an increased production of IL-6, IL-1β, and NO in the culture media of HGF-1 cells compared to LPS from *P. gingivalis*. The same pattern was previously reported on the ESK-1 mouse gingival fibroblast cell line [97]. However, the inflammatory response induced by LPS stimulation depends on the cell-LPS binding interaction [97]. Furthermore, various studies have shown that there are major differences between the lipid A structure of *P. gingivalis* and *E. coli* LPS, which lead to marked differences in their endotoxicity [98]. The toll-like receptor (TLR) signaling pathway plays a crucial role in innate immune response activation following bacterial infection [99], together with the signal transduction by the pattern recognition receptor (PRR) cluster of differentiation 14 (CD14). CD14 is highly expressed by HGF cells after *P. gingivalis* LPS stimulation due to the CD14 recognition of LPS via the lipid A moiety [100]. Also, it was previously reported that the *E. coli* LPS increases the expression level of TLR-4 in human dental pulp stem cells, while the expression level of TLR-2 remained at the level of the negative control (untreated cells). Instead, *P. gingivalis* LPS did not affect the TLR-4 and TLR-2 expression levels. Therefore, the level of IL-6 was significantly higher following the *E. coli* LPS stimulation [101]. In the current study, the Se50BNCSFa nanoformulation proved to significantly reduce the levels of IL-6 and IL-1β released by HGF-1 gingival fibroblasts when stimulated with 1 µg/mL LPS from *P. gingivalis* or *E. coli*. The extracellular release of TNF-α was significantly lower after 24 h of cell incubation with Se50BNCSFa and 1 µg/mL LPS from *E. coli* when compared to the positive control for inflammation. Instead, there was only a slight reduction without statistical significance in the amount of TNF-α in the culture media when the cells were treated with Se50BNCSFa and 1 µg/mL LPS from *P. gingivalis* by comparison with the positive control for inflammation. The same pattern recorded for TNF-α was noticed in the case of NO production. Ferulic acid grafted onto chitooligosaccharides was reported to decrease the levels of IL-6, TNF-α, and IL-1β synthesized by murine macrophage cells, as well as inhibit NO production [102]. SeNPs, especially biogenic ones, can also lead to inflammation resolution, mainly due to their antioxidant activity [103]. In another study, Se-methylseleno-L-cysteine inhibited the RNA expression of iNOS, TNF-α, IL-1β, IL-6, etc., in murine RAW 264.7 macrophages. The proposed mechanism was the inhibition of the major signaling pathways that contribute to the regulation of pro-inflammatory mediator levels [104].

There are several studies that highlight the increased potential of *Lactobacillus* sp. to inhibit the growth of pathogenic bacteria. *L. reuteri* is an obligate heterofermentative species that produces lactic acid [105]. Glucose is used as a carbon source in order to produce pyruvate by glycolysis, and subsequently, lactic acid is produced under the action of lactate dehydrogenase. Glucose can also be decomposed to ethanol via the phosphoketolase (PK) pathway in heterolactic fermentation [106]. It was found that *L. reuteri* can produce H_2_O_2_ from lactic acid and ammonia from arginine [105,107], which can lead to the removal of the free lactic acid level, increasing the pH, and combating the acid-tolerance response system, which is crucial for different pathogenic bacteria [105]. Moreover, *L. reuteri* produces reuterin, an organic compound that acts as an antibiotic. *L. reuteri* inhibited the growth of several pathogenic oral bacteria: *S. mutans*, or Gram-negative bacteria like *P. gingivalis*, and *Fusobacterium nucleatum* [108]. *L. salivarius* is a homofermentative species that can produce only lactic acid through the carbohydrate consumption [109]. It has been reported that *L. salivarius* has increased adherence to human gingival fibroblast cells when compared to other lactobacilli strains, but it also presented antibacterial activity against two oral pathogens, i.e., *Aggregatibacter actinomycetemcomitans* and *Actinomyces naeslundii* [110]. Therefore, probiotics can promote the homeostasis of the oral microbiota.

Se50BNCSFa stimulated the growth of both *L. reuteri* and *L. salivarius* compared with control after 24 h. Prebiotics are food components that act as a growth substrate for probiotic microorganisms and also contribute to the health of the host organism [111]. Prebiotics include polyphenols and different types of carbohydrates: fructans (e.g., fructooligosaccharides (FOS) and inulin), galactans (e.g., galactooligosaccharides (GOS)), mannans (e.g., mannan oligosaccharides (MOS)), and xylans (e.g., xylooligosaccharides (XOS)) [112]. Cellulose is widely known as a dietary fiber, and its intake can modulate the gut microbiota [113]. Additionally, cellulose with nanometric scale size has increased bioavailability, promoting gut microbiota growth and increasing the production of short-chain fatty acids (SCFAs) [114]. There are also studies about the prebiotic activity of chitosan, oligochitosan, and chitoligosaccharides (COS) [115,116,117]. It was found that water-soluble chitosan obtained from crab shell chitin had a significant impact on the intestinal microbiota of mice by increasing the microbial diversity [116]. Another study reported that COS increased the population of lactobacilli in the mouse bowel and slightly elevated the level of cecal SCFAs [117]. Polyphenols have direct and indirect/mediated prebiotic effects. The direct prebiotic effects involve the metabolization of polyphenols by probiotic microorganisms, especially those from the genus *Lactobacillus* [118]. Indirect mechanisms involve the stimulation of mucin secretion and, consequently, the proliferation of probiotic bacteria that breakdown the intestinal mucus (e.g., *Akkermansia* spp.) [119]. It was reported that ferulic acid from rice bran dietary fiber is metabolized by the gut microbiota into vanillin and caffeic acid, with the latter being converted into hydroxyphenylpropionic acid. Moreover, the phenolics released have increased *Lactobacillus* spp., *Akkermansia muciniphila*, and *Faecalibacterium prausnitzii* populations [120]. Molan et al. investigated the effect of Se-enriched green tea (1.4 mg Se/kg; 1–5 µg Se/mL), and the results highlighted a significant increase in the lactobacilli and bifidobacteria growth in comparison with the non-supplemented green tea [121]. Moreover, in another study by Molan et al., the oral administration of Se-enriched green tea significantly increased the lactobacilli and bifidobacteria populations in the cecal microbiota of the rats [122]. In another study, it was shown that Se-enriched sauerkraut (0.3 mg Na_2_SeO_3_/kg fresh cabbage) had higher lactic acid bacteria counts than the control without Se [123]. In bacteria, including lactobacilli, Se is an essential trace element needed, for example, for the synthesis of the 21st amino acid selenocysteine, further incorporated in various selenoproteins such as oxidoreductases, as well as for incorporation in Se-dependent molybdenum hydroxylases (SDMHs), although SDMHs have not been reported in *Lactobacillus* sp. yet [124,125,126]. Our hydrogel formulations contain various types of these already proven prebiotics, which explains the growth stimulation of *L. reuteri* and *L. salivarius* compared with control.

There were nevertheless some differences in behavior between the two strains. SeNPsK inhibited *L. reuteri* and stimulated *L. salivarius* at the dose used. There are not many studies to investigate the effects of Se, especially in the form of SeNPs, on *Lactobacillus* sp. The available reported research indicates that there are significant differences in Se responses between various lactobacilli [127,128,129,130]. As with other organisms, the effects depend on the dose and can range from stimulation at low doses to inhibition at higher doses [131,132,133]. The line between the beneficial, tolerated, and inhibitory Se doses seems to be *Lactobacillus* strain dependent, with the exact cause being unclear. The effects greatly depend on the Se form as well, with SeNPs being more tolerated than salt (selenate, selenite) forms [133,134]. Moreover, it was shown that the type of Se species has an effect on the profile of expressed proteins [134]. Our results confirm previous observations of differences between the responses of *Lactobacillus* strains to selenium, although we could find no study to investigate these differences with Se applied as SeNPs. For the first time, to the best of our knowledge, our study indicates that strain dependence occurs even in the response to SeNPs. Another difference was observed with respect to the effect of hydrogel over time. The effect was maintained even after 48 h and 72 h of incubation in the case of *L. reuteri*, but not in the case of *L. salivarius*. Whether the differences observed depend on the different metabolisms of the two strains, such as the level of necessary Se and hetero- versus homofermentation, needs to be further investigated, as no systematic study is available in this respect to the best of our knowledge.

Among the culprits responsible for the most implant-related infections are pathogenic microorganisms or opportunistic pathogens like *S. aureus*, *E. coli*, *C. albicans*, *P. aeruginosa* [135], and *B. cereus* [136]. The antimicrobial activity of ferulic acid-grafted chitosan against pathogenic microorganisms has been previously proven [28,76,78]. The Gram-positive bacteria (e.g., *S. aureus*, *B. cereus*) present a thick cell wall rich in peptidoglycans and theicoic acids, which confer on them both rigidity and elasticity. In contrast, the cell wall of Gram-negative bacteria (e.g., *E. coli*, *P. aeruginosa*) is composed of a thin layer of peptidoglycans and an outer membrane [137]. Even if CS was proved to be effective against a wide range of Gram-positive and Gram-negative bacteria, it was found that its antibacterial efficiency is increased in phenolic acid-CS conjugates not only because of the additive effect that occurs through the interaction of the two compounds but also because of the formation of 6-*N*-substituted chitosan derivatives [138]. Moreover, the bacterial membrane possesses a negative surface charge under physiological conditions. By analyzing the zeta potential following exposure of *S. aureus*, *E. coli*, *P. aeruginosa*, and *L. monocytogenes* to different concentrations of gallic acid and ferulic acid (100–1000 µg/mL), it was observed that bacterial cells become less electronegative [139]. Furthermore, the influx of phenolic acids through the bacterial membrane by passive diffusion due to their predominant lipophilic character leads to membrane disruption and loss of cell constituents (proteins, nucleic acids, potassium, phosphate, etc.) [139,140]. Zhang et al. proposed an antibacterial mechanism for SeNPs synthesized by Providencia sp. DCX. After adsorption of biogenic SeNPs to the bacterial cell surface via electrostatic interactions, the level of ROS increases, leading to oxidative stress and further membrane disruption with loss of cellular constituents, as well as DNA damage and transmembrane electron transport dysfunctions [141].

In the current study, the semiquantitative screening of the antimicrobial activity showed an inhibitory activity of 13.30 ± 0.10 mm inhibition zone against *E. coli* and 9.41 ± 0.04 mm inhibition zone against *S. aureus*. A study conducted by Yang et al. [76] showed that ferulic acid-grafted chitosan synthesized by enzyme-catalyzed grafting exhibited antibacterial properties against various strains, including *E. coli* ATCC 25922, *S. aureus* ATCC 25923, and *B. subtilis* ATCC 168. By employing the Oxford Cup method, the authors obtained results for the inhibition zone ranging from 17.03 ± 0.20 mm (against *E. coli*) to 17.64 ± 0.32 mm (against *S. aureus*) by using 200 µL of 1 mg/mL ferulic acid-grafted chitosan. In another study [26], the researchers developed a bacterial cellulose–chitosan–ferulic acid composite (BCF) based on bacterial cellulose (BC) and chitosan grafted with ferulic acid using the free radical-mediated grafting method at a molar ratio of 1:0.1. The obtained results showed that the diameter of the inhibition zone of the BCF film was 13.3 ± 0.10 mm and 13.2 ± 0.10 mm against *E. coli* and *S. aureus*, respectively, after 24 h of incubation. In the present study, at 12 h post-treatment, Se50BNCSFa induced an inhibition of *S. aureus* growth of about 20% at the highest microbial density. Its antimicrobial potential increased by decreasing the microbial density. At 24 h post-treatment, the antimicrobial activity of Se50BNCSFa slightly decreased at the highest microbial density in comparison with the antimicrobial activity recorded at 12 h post-treatment, but it increased at lower microbial densities. All in all, Se50BNCSFa showed great potential to prevent *S. aureus* growth. The semiquantitative screening of the antimicrobial activity showed an inhibitory activity of 8.389 ± 0.181 mm against *B. cereus* and 8.549 ± 0.046 mm against *P. aeruginosa*. Other authors also reported the antibacterial activity of ferulic acid conjugates against *P. aeruginosa* [48,142]. The antimicrobial activity of Se50BNCSFa against *B. cereus* is significantly higher when compared to the 50BNCSFa hydrogel at 12 h and 24 h post-treatment, which highlights the increased antimicrobial effect of SeNPsK against *B. cereus*. The quantitative screening indicated that 50BNCSFa possesses a similar antimicrobial potential as Se50BNCSFa against *P. aeruginosa*, with the hydrogel nanoformulations being slightly effective and only at 12 h post-treatment.

By dropwise application of 25 µL of SeBNCSFa on the surface of SDA previously seeded with *C. albicans*, the diameter of the inhibition zone of the microbial growth was 12.45 ± 0.65 mm. Another study revealed by XTT assay that at 1 × 10^7^ microbial cells/mL, the encapsulation of ferulic acid into chitosan nanoparticles by the ionic gelation method led to a decrease in the metabolic activity of *C. albicans* up to 22.5%, while the non-encapsulated chitosan nanoparticles and ferulic acid reduced the cell viability only up to 88% and 63%, respectively, indicating an increased bioactivity of the ferulic acid encapsulated chitosan nanoparticles after 24 h of incubation. Moreover, in the aforementioned study, fluconazole was inefficient up to 128 µg/mL [143]. In comparison, the antimicrobial activity of Se50BNCSFa against *C. albicans* ranged between 10 and 75% at 12 h post-treatment, depending on the inoculated microbial density. After 24 h, Se50BNCSFa showed an antimicrobial potential of approximately 18% at the lowest microbial density, which shows the potential of SeNPsK-enriched nanoformulation to prevent microbial infection even at a low dose.

## 5. Conclusions

A hydrogel nanoformulation enriched with SeNPs from Kombucha fermentation was developed based on CSFa(+) prepared by the free radical-mediated grafting method of chitosan with FA and BNC with Iβ-cellulose-reinforced structure. The grafting degree was approximately 1.78 ± 0.07% (17.8% efficiency) considering single-site grafting per chitosan monomer. The grafting was evidenced by FTIR spectroscopy through the shifts of amide I, II, and III bands to lower wavenumbers and also by a new small band around 1753 cm^−1^ characteristic of ester bonds. X-ray diffraction confirmed the grafting through a decreased crystallinity of the grafted chitosan, together with two new diffraction peaks around the 2θ angles 9.12° and 13.02° related to ferulic acid. The grafting was also confirmed by thermogravimetry as a convoluted derivative curve of the weight loss of grafted ferulic acid with the chitosan backbone at lower decomposition temperatures than the ones for neat compounds. The SeBNCSFa hydrogel, characterized through the same analytical methods, appeared as a particular nano-blend of highly rearranged molecules, having multiple hydrogen bonds and ionic interactions between the hydroxyl and carboxyl groups of ferulic acid and the hydroxyl, acetyl, and amino groups of nanocellulose and chitosan. The hydrogel adhesion to a titanium surface tested by axial rheology evidenced an adhesion force of 0.105 N and an 8 s adhesion time at maximum axial force. The lyophilized form of the hydrogel also appears as a nano-structured, interconnected blend with a particular diffraction pattern and thermal decomposition behavior. The Se50BNCSFa treatment presented a high degree of cytocompatibility, with the potential to stimulate cell proliferation up to 107.7 ± 2.16% of C−. A significant decrease in the amount of intracellular ROS was observed, the level being 4-times lower compared to the positive control (ROS-inducing agent). The level of pro-inflammatory mediators released in the culture media following the cell stimulation with 1 µg/mL LPS from *P. gingivalis* and *E. coli* was decreased in the presence of Se50BNCSFa nanoformulation. A significant potential for preventing microbial growth was observed at 12 h and 24 h post-treatment. The antioxidant and antibacterial activities, as well as the cytocompatibility behavior of SeBNCSFa, are higher in comparison with our previous nanoformulation based on a CS-BNC matrix enriched with SeNPs phytosynthesized by sea buckthorn leaf extract. This suggests an increased biological activity of CSFa(+) conjugate in comparison with non-grafted chitosan, as well as the unique properties of SeNPsK that result from the biosynthesis process.

The SeBNCFa nanoformulation may be considered a potential candidate for improving the efficacy and safety of dental implants due to its high degree of cytocompatibility, antioxidant, antimicrobial, and anti-inflammatory activities, as well as its adhesion to the titanium surface.

## Data Availability

The original contributions presented in the study are included in the article/Appendix A, further inquiries can be directed to the corresponding authors.

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
