# Peer review of "Bioactive Hydrogel Formulation Based on Ferulic Acid-Grafted Nano-Chitosan and Bacterial Nanocellulose Enriched with Selenium Nanoparticles from Kombucha Fermentation"

_jfb, 2024, doi:10.3390/jfb15070202_

Round 1

Reviewer 1 Report

Comments and Suggestions for Authors

1. The article mentions that the application dosage range of selenium nanoparticles is relatively narrow, which is considered a less than ideal characteristic compared to silver nanoparticles, gold nanoparticles, and others that have a broader range of dosages for applications. In comparison to other metal nanoparticles, what advantages does selenium nanoparticles offer?

2. After analyzing the XRD results in the main text (Page 9, Line 392), it is recommended to create a new paragraph to discuss the TGA and DTG experimental results. This restructuring will enhance the clarity of the article's structure.

3. The article correctly labels Figure 4a and Figure 4b when analyzing the TGA and DTG results in the main text. However, there is a missing reference to Figure 4c and Figure 4d in the subsequent text. It is advisable to include the appropriate references to Figure 4c and Figure 4d to maintain consistency and ensure that all figures are properly acknowledged and discussed in the text.

4. The caption for Figure 4 contains an error. The correct caption should be "(d) TGA and DTG of SeBNCFa" instead of "(c) TGA and DTG of SeBNCFa". Please make sure to update the caption accordingly to maintain accuracy in the figure labeling.

5. The main text of the article lacks labeling or explanatory notes for Figure 8 and Figure 6f. It is necessary to provide appropriate captions or explanations for these figures to ensure clarity and completeness in the presentation of the research findings.

6. The article only setting the SeBNCFa hydrogel group in the rheological adhesion experiment raises a concern. Including additional control groups in the experiment is recommended to enable a comparative analysis and effectively demonstrate the adhesive properties of the designed hydrogel on titanium plates.

7. One crucial aspect of evaluating a hydrogel system is its swelling and water retention properties, which are essential for assessing the stability of the hydrogel and drug release. The article lacks relevant experiments to investigate these characteristics.

Comments on the Quality of English Language

The quality of English language in the document is fine.

Author Response

Point 1. The article mentions that the application dosage range of selenium nanoparticles is relatively narrow, which is considered a less than ideal characteristic compared to silver nanoparticles, gold nanoparticles, and others that have a broader range of dosages for applications. In comparison to other metal nanoparticles, what advantages does selenium nanoparticles offer?

Response 1.  We corrected the text, it was about the dosage range of selenium, not SeNPs. SeNPs are known to have lower toxicity than other Se forms. We added in the Introduction more explanations to justify the use of SeNPs (lines 92-100). We selected Se because of its additional roles to cellular function compared with other NPs which generally provide only antimicrobial and antioxidant activities. SeNPs is a good and less toxic source of Se compared with Se salts and have antimicrobial and antioxidant activities as well.

Point 2. After analyzing the XRD results in the main text (Page 9, Line 392), it is recommended to create a new paragraph to discuss the TGA and DTG experimental results. This restructuring will enhance the clarity of the article's structure.

Response 2. We created a new paragraph. Thank you.

Point 3. The article correctly labels Figure 4a and Figure 4b when analyzing the TGA and DTG results in the main text. However, there is a missing reference to Figure 4c and Figure 4d in the subsequent text. It is advisable to include the appropriate references to Figure 4c and Figure 4d to maintain consistency and ensure that all figures are properly acknowledged and discussed in the text.

Response 3. Thank you very much for the observation. We inserted them in the main text.

Point 4. The caption for Figure 4 contains an error. The correct caption should be "(d) TGA and DTG of SeBNCFa" instead of "(c) TGA and DTG of SeBNCFa". Please make sure to update the caption accordingly to maintain accuracy in the figure labeling.

Response 4. Thank you. We modified.

Point 5. The main text of the article lacks labeling or explanatory notes for Figure 8 and Figure 6f. It is necessary to provide appropriate captions or explanations for these figures to ensure clarity and completeness in the presentation of the research findings.

Response 5. Thank you very much. We inserted the explanatory notes for Figure 8 and 6f (now 7f).

Point 6. The article only setting the SeBNCFa hydrogel group in the rheological adhesion experiment raises a concern. Including additional control groups in the experiment is recommended to enable a comparative analysis and effectively demonstrate the adhesive properties of the designed hydrogel on titanium plates.

Response 6. Thank you for the suggestion. We inserted the rheological analysis in the main text (Figure 5, lines 657-716) and in the Supplementary Materials: the rheological tests for the components of the SeBNCSFa formulation, as well as for the initial CS. There is no available and comparable standard to include in the analysis, but we included comparison with formulations that are more specific for this purpose (1154-1169). We need to mention that the main targeted property of our hydrogel was not high adhesiveness to Ti, as the mode of application is not as an adhesive, but as an adjuvant for inflammation resolution and better tissue regeneration, as well as pathogen growth inhibition. Our hydrogel has a low viscosity and therefore the adhesion to Ti is lower than other formulations, but it is higher than pathogen adhesion previously reported, which can help to inhibit pathogen adhesion, besides its growth (lines 1169-1172).

Point 7. One crucial aspect of evaluating a hydrogel system is its swelling and water retention properties, which are essential for assessing the stability of the hydrogel and drug release. The article lacks relevant experiments to investigate these characteristics.

Response 7. Thank you for your observation. We could not perform swelling and water retention, because our hydrogel is based only on biopolymers and disperses into water relatively fast, as it is not cross-linked / reticulated. The purpose was that all components of the hydrogel (including the biopolymers) contribute to various biological aspects involved, therefore it is a more atypical hydrogel.

Reviewer 2 Report

Comments and Suggestions for Authors

This manuscript prepared a hydrogel formulation (SeBNCFa) based on ferulicacid-grafted chitosan (CFa) and bacterial nanocellulose (BNC) enriched with SeNPs from Kombucha fermentation (SeNPsK). The SeBNCFa was characterized by SEM, TEM, XRD, and TGA. Moreover, the cell viability, antioxidant property, and anti-inflammatory behavior of hydrogel were performed. However, lacking of some data is remaining issues of this manuscript. Therefore, I suggested it should be published after minor revision as the case standards. Here are the questions and suggestions about the manuscript.

1.      The authors claimed that hydrogel could adhere to titanium surface. However, no related data was presented.

2.      The rheology behavior and swelling behavior of hydrogel were recommended to be added.

3.      How about the SeNPs release behavior? It would be better to add the SeNPs release behavior of hydrogel.

4.      Apart from the diameter of the inhibition zone picture, the bactericidal picture of hydrogel was suggested to be added.

5.      What’s the antibacterial mechanism of hydrogel?

6.      Some related publications were suggested to be added. For examples, https://doi.org/10.1007/s12274-022-5129-1 and https://doi.org/10.1016/j.ajps.2022.01.001

Comments on the Quality of English Language

English is good. 

Author Response

This manuscript prepared a hydrogel formulation (SeBNCFa) based on ferulic acid-grafted chitosan (CFa) and bacterial nanocellulose (BNC) enriched with SeNPs from Kombucha fermentation (SeNPsK). The SeBNCFa was characterized by SEM, TEM, XRD, and TGA. Moreover, the cell viability, antioxidant property, and anti-inflammatory behavior of hydrogel were performed. However, lacking of some data is remaining issues of this manuscript. Therefore, I suggested it should be published after minor revision as the case standards. Here are the questions and suggestions about the manuscript.

Point 1. The authors claimed that hydrogel could adhere to titanium surface. However, no related data was presented.

Response 1. Thank you for the suggestion. We inserted the adhesion tests in axial mode on Quartz (the standard surface of our rheometer) and on Ti for the SeBNCFa matrix (BNC and CSFa), SeBNCFa, and initial CS (Figure 5 and Supplementary Materials, lines 699-716).

Point 2. The rheology behavior and swelling behavior of hydrogel were recommended to be added.

Response 2. We inserted the rheological analysis (in oscillatory and flow mode) in the main text (Figure 5, lines 657-698) and in the Supplementary Materials. The swelling behavior could not be performed as the hydrogel solubilizes relatively fast in water.

Point 3. How about the SeNPs release behavior? It would be better to add the SeNPs release behavior of hydrogel.

Response 3. Thank you for your observation. We cannot perform SeNPs release behavior because our hydrogel is not a cross-linked/reticulated one. Being formulated only on biopolymers, the hydrogel easily disperses completely in solution and the observed effects come from all components, not just SeNPs.

Point 4. Apart from the diameter of the inhibition zone picture, the bactericidal picture of hydrogel was suggested to be added.

Response 4. Thank you for your suggestion. We included the bactericidal effect images in the Supplementary Materials (Table S7) and commented in the text (1011-1013) and the method is described at lines 466-472.

Point 5. What’s the antibacterial mechanism of hydrogel?

Response 5. Thank you. We included some information in the Discussion section (lines 1354-1373). We performed additionally prebiotic activity on two Lactobacillus strains (lines 887-911)

Point 6. Some related publications were suggested to be added. For examples, https://doi.org/10.1007/s12274-022-5129-1 and https://doi.org/10.1016/j.ajps.2022.01.001

Response 6. Thank you for your recommendations. We included them in the Introduction section (second paragraph).

Reviewer 3 Report

Comments and Suggestions for Authors

The authors of the manuscript under review prepared a bioactive hydrogel based on chitosan grafted with ferulic acid and bacterial nanocellulose enriched with selenium nanoparticles. It is intended for use as an adjuvant for oral implant integration. The volume of experimental work is very large, the methods are varied, and the results are promising. However, the manuscript suffers from a number of shortcomings that should be eliminated before publication.

The main polymer, chitosan, is not fully characterized. Its viscosity is given (line 100; why not the intrinsic viscosity?), but the average molecular weight appears only in line 126, and the degree of deacetylation is not present. But the authors carry out calculations, evaluate the degree and efficiency of grafting, and knowledge of the exact average molecular weight of one chitosan unit is mandatory. E.g., in lines 129 and 130 we read: “…in a molar ratio of chitosan:FA of 1:0.1”—do you mean moles of chitosan monomer units with only amino groups, or with acetamide groups too? Line 145: “…based on their molecular weights”—same issue. Line 344: «C=O, N-H and C-N vibrations in residual acetyl groups”—that is, the authors know about the presence of these groups, but do not try to estimate their quantity.

The authors repeat several times throughout the text: “fungal chitosan”, “chitosan from crab shells”, etc. Strictly speaking, it is chitin which is obtained from natural raw materials, and chitosan is further obtained therefrom, the properties of which, therefore, depend not only on the source (nature) of the raw material, but on deacetylation conditions as well. By the way, the designation “CFa” does not seem appropriate; the letter “C” may mean chitin, carbon, “sterility control”, and “cytotoxicity negative control (C-)”. Besides, the authors themselves (line 60 onwards) designate chitosan as “CS”.

The term “monomer” is used incorrectly. Line 34: “total chitosan monomers”, line 149: “grafted monomers”, line 146: “the percent of grafted monomers”—what is meant, chitosan monomer units or ferulic acid molecules? Each such molecule contains a double bond and can, in principle, enter into a chain polymerization reaction. The authors should have provided a scheme of the grafting reaction, as, for example, in Ref. [25] (Woranuch and Yoksan, 2013). If grafting occurs by the amino groups of chitosan with the participation of the carboxyl group of ferulic acid, has it been taken into account that acetic acid used to dissolve chitosan can react in a similar way? And were the existing acetamide groups left over from chitin taken into account?

In Subsection 2.2, the authors, in support of their grafting technique, refer to article [42] (Curcio et al., 2009). However, in this work chitosan is grafted with gallic (not ferulic) acid and catechin. At the same time, there are works on grafting ferulic acid onto chitosan. The authors themselves write (lines 882 and 833): “The antimicrobial activity of ferulic acid-grafted chitosan against pathogenic microorganisms has been previously proven [67, 69, 70]”. Why was the grafting technique not taken from these works? Moreover, these and similar papers, references to which are replete with Section 4 “Discussion”, should have been cited in Section 1 “Introduction” when justifying the choice of research objects and when determining the prototype, i.e. an object known from the literature which has the greatest similarity to the one invented by the authors. And in the “Conclusion” section, the authors should compare their hydrogels with this prototype—not just summarize the properties studied.

The volume of the Abstract exceeds the maximum allowable (~250 words vs. 200). In Subsection 2.1 “Materials,” the chemicals used are listed in one long phrase (lines 99–122), with individual substances, drugs, and commercial specialty mixtures mixed together. It is desirable to provide separate lists and indicate purity (grade) for individual substances. Note in passing that very long paragraphs (such as those on page 9 or in Section 4) make them difficult to read/understand.

In Subsection 2.4.4, the authors describe the XRD sample analysis procedure. It is known, however, that the degree of crystallinity of polymers depends on crystallization conditions, which are not described. Drying of the grafted chitosan should have been described in Subsection 2.2, especially since further weighing was apparently carried out to prepare solutions with precise concentrations. Did the authors take moisture content into account?

Lines 381–383: “The grafted chitosan CFa shows a reduced crystallinity degree of 37% and three peaks around 9.12°, 13.02° and 19.84°, the first two small peaks being correlated with the peaks at 9.02° and 12.82° from ferulic acid”. I.e., the crystal structure of at least some regions of CFa is similar to that of ferulic acid. But how can this be? The ferulic acid content is quite low (grafting efficiency of ~1.5%), how could acid units occur in such quantities to form a crystal lattice? Perhaps the excess acid was not completely removed by dialysis, and crystallized Fa grains were present in the dry sample?

The grafting efficiency (Eq. (1)) is calculated incorrectly. “The molar ratio of chitosan:FA was 1:0.1” (line 129), so grafting all the acid (1:10) would be 100% effective. If instead 1.5% of chitosan units were grafted, then the efficiency is 15%, whilst 1.5% is the grafting degree. In addition, since the work is of an applied nature, it is the degree of grafting (material composition) that is important, whose target value should be achieved at any efficiency.

Line 314: “16.70 mg ± 0.39 ferulic acid (FA) /g chitosan (approximately 13.7 molar ratio FA:chitosan)”. Do the authors mean that 13.7 molecules of ferulic acid were grafted onto one monomeric unit of chitosan? Maybe it’s a millimolar ratio?

Non-round numbers are striking: “0.432 g of ascorbic acid”, “12.04 mg/mL”, etc. What was the reason for this choice of masses and concentrations? If round numbers of moles (molarities) were used, this should be noted. And sometimes numbers are given with clearly excessive precision (for example, “10.591 ± 0.235” in Table 1). The explanation “titanium (Ti)” (line 795) is redundant—the reader is probably familiar with the Periodic Table.

A number of abbreviations are defined many times. NDBNC (lines 60 and 155). TGA (lines 200 and 419). SeBNCFa (lines 163, 325, 334, 368, 422, 441, and 667). BNCFa (lines 154, 434, 440, 466, 499, 780, and 815).

The manuscript can be recommended for publication after eliminating the noted shortcomings.

Author Response

Point 1. The authors of the manuscript under review prepared a bioactive hydrogel based on chitosan grafted with ferulic acid and bacterial nanocellulose enriched with selenium nanoparticles. It is intended for use as an adjuvant for oral implant integration. The volume of experimental work is very large, the methods are varied, and the results are promising.

Response 1. Thank you very much.

Point 2. However, the manuscript suffers from a number of shortcomings that should be eliminated before publication. The main polymer, chitosan, is not fully characterized. Its viscosity is given (line 100; why not the intrinsic viscosity?), but the average molecular weight appears only in line 126, and the degree of deacetylation is not present. But the authors carry out calculations, evaluate the degree and efficiency of grafting, and knowledge of the exact average molecular weight of one chitosan unit is mandatory. E.g., in lines 129 and 130 we read: “…in a molar ratio of chitosan:FA of 1:0.1”—do you mean moles of chitosan monomer units with only amino groups, or with acetamide groups too? Line 145: “…based on their molecular weights”—same issue. Line 344: «C=O, N-H and C-N vibrations in residual acetyl groups”—that is, the authors know about the presence of these groups, but do not try to estimate their quantity.

Response 2. The viscosity given was the one mentioned by the manufacturer. We added the measurements of the intrinsic viscosity, molecular weight (we corrected the previous value after careful re-evaluation), and degree of deacetylation, see lines 197-226 for the methods and lines 492-497 for the results. We used these values to recalculate the average molecular weight and the degree of grafting. The molar ratio refers to moles of chitosan monomer units considering both acetylated and free amino groups (average molecular weight). We added this detail in the text.

We additionally performed DLS and Zeta potential analysis of initial, activated and grafted chitosan (lines 515-529, Figures and Tables in Supplementary Materials).

Point 3. The authors repeat several times throughout the text: “fungal chitosan”, “chitosan from crab shells”, etc. Strictly speaking, it is chitin which is obtained from natural raw materials, and chitosan is further obtained therefrom, the properties of which, therefore, depend not only on the source (nature) of the raw material, but on deacetylation conditions as well. By the way, the designation “CFa” does not seem appropriate; the letter “C” may mean chitin, carbon, “sterility control”, and “cytotoxicity negative control (C-)”. Besides, the authors themselves (line 60 onwards) designate chitosan as “CS”.

Response 3. Thank you for your suggestions. We modified the expression. We changed C to CS, so CFa became CSFa, as well as the rest of the abbreviations (i.e., BNCSFa, SeBNCSFa, etc.), which were changed in the figures as well.

Point 4. The term “monomer” is used incorrectly. Line 34: “total chitosan monomers”, line 149: “grafted monomers”, line 146: “the percent of grafted monomers”—what is meant, chitosan monomer units or ferulic acid molecules? Each such molecule contains a double bond and can, in principle, enter into a chain polymerization reaction. The authors should have provided a scheme of the grafting reaction, as, for example, in Ref. [25] (Woranuch and Yoksan, 2013). If grafting occurs by the amino groups of chitosan with the participation of the carboxyl group of ferulic acid, has it been taken into account that acetic acid used to dissolve chitosan can react in a similar way? And were the existing acetamide groups left over from chitin taken into account?

Response 4. The monomers were meant to refer everywhere to chitosan monomeric units (acetylated and/or deacetylated) present in solution and calculated from the mass and average molecular weight of monomer unit according to the deacetylation degree (independent of polymer length). We added everywhere “chitosan” and “unit”, to be clearer. We added a scheme of the grafting reaction in the Discussion section.

We proved by NMR (Supplementary materials and lines 493-497) that the degree of acetylation was the same after grafting. The acetamide groups were taken into account when calculating the average molecular weight of the monomer unit.

Point 5. In Subsection 2.2, the authors, in support of their grafting technique, refer to article [42] (Curcio et al., 2009). However, in this work chitosan is grafted with gallic (not ferulic) acid and catechin. At the same time, there are works on grafting ferulic acid onto chitosan. The authors themselves write (lines 882 and 833): “The antimicrobial activity of ferulic acid-grafted chitosan against pathogenic microorganisms has been previously proven [67, 69, 70]”. Why was the grafting technique not taken from these works? Moreover, these and similar papers, references to which are replete with Section 4 “Discussion”, should have been cited in Section 1 “Introduction” when justifying the choice of research objects and when determining the prototype, i.e. an object known from the literature which has the greatest similarity to the one invented by the authors. And in the “Conclusion” section, the authors should compare their hydrogels with this prototype—not just summarize the properties studied.

Response 5. Thank you for your observation. We corrected the reference, the protocol was not taken from that reference, but from a paper where they grafted ferulic acid as well, it was a mistake.

We have added in the Introduction the following text: “Ferulic acid-grafted chitosan was proved to be the most effective against several Gram-positive and Gram-negative bacteria, including methicillin-resistant Staphylococcus aureus strains, in comparison with other HCA-CS conjugates, i.e., caffeic acid-grafted chitosan and sinapic-acid grafted chitosan” and “Moreover, we investigated in a previous study the biological activity of a hydrogel nanoformulation with a CS-BNC matrix embedding SeNPs phytosynthesized by sea buckthorn leaf extract”. We modified the last phrase of the Introduction to highlight better the aim in this context: “The aim of this study was focused on the development of a hydrogel formulation based on ferulic acid-grafted chitosan and NDBNC enriched with SeNPs from Kombucha fermentation (SeNPsK) with increased cytocompatibility, antioxidant, and antibacterial activities compared with the previous hydrogel, as well as high anti-inflammatory effect.

We added in the Conclusion section the following text: “The antioxidant and antibacterial activities, as well as the cytocompatibility behavior of SeBNCSFa are higher in comparison with our previous nanoformulation based on a CS-BNC matrix enriched with SeNPs phytosynthesized by sea buckthorn leaf extract. This suggests an increased biological activity of CSFa(+) conjugate in comparison with non-grafted chitosan, as well as the unique properties of SeNPsK which result from the biosynthesis process.” (lines 1429-1437)

Point 6. The volume of the Abstract exceeds the maximum allowable (~250 words vs. 200). In Subsection 2.1 “Materials,” the chemicals used are listed in one long phrase (lines 99–122), with individual substances, drugs, and commercial specialty mixtures mixed together. It is desirable to provide separate lists and indicate purity (grade) for individual substances. Note in passing that very long paragraphs (such as those on page 9 or in Section 4) make them difficult to read/understand.

Response 6. We reduced the Abstract section to 200 words. We also separated the substances, commercial mixtures depending on different activities.

Point 7. In Subsection 2.4.4, the authors describe the XRD sample analysis procedure. It is known, however, that the degree of crystallinity of polymers depends on crystallization conditions, which are not described. Drying of the grafted chitosan should have been described in Subsection 2.2, especially since further weighing was apparently carried out to prepare solutions with precise concentrations. Did the authors take moisture content into account?

Response 7. We included the crystallization conditions of all samples by freeze-drying in the Subsection 2.2. and 2.3, which were the same for the XRD analysis. After the freeze-drying process, the samples were kept in the desiccator and analyzed in less than 24h for all the physical-chemical analysis. The water content of the samples after the freeze-drying process was considered structural water or crystallization water, as the TGA weight loss at 105°C in N2, of around 8.89-10.17%, evidenced.

Point 8. Lines 381–383: “The grafted chitosan CFa shows a reduced crystallinity degree of 37% and three peaks around 9.12°, 13.02° and 19.84°, the first two small peaks being correlated with the peaks at 9.02° and 12.82° from ferulic acid”. I.e., the crystal structure of at least some regions of CFa is similar to that of ferulic acid. But how can this be? The ferulic acid content is quite low (grafting efficiency of ~1.5%), how could acid units occur in such quantities to form a crystal lattice? Perhaps the excess acid was not completely removed by dialysis, and crystallized Fa grains were present in the dry sample?

Response 8. We did not want to suggest that the two small diffraction peaks at 9.12° and 13.02° belong to non-grafted crystallized ferulic acid. We did not detect any free ferulic acid after dialysis. We wanted to suggest that the grafting with ferulic acid induces some conformational changes within chitosan that results in its diffractogram transformation. We changed in the main text as follows: “The first two small peaks could indicate a new arrangement of the first chitosan peak at 11.08° upon grafting with ferulic acid.” (lines 605-608).

Point 9. The grafting efficiency (Eq. (1)) is calculated incorrectly. “The molar ratio of chitosan:FA was 1:0.1” (line 129), so grafting all the acid (1:10) would be 100% effective. If instead 1.5% of chitosan units were grafted, then the efficiency is 15%, whilst 1.5% is the grafting degree. In addition, since the work is of an applied nature, it is the degree of grafting (material composition) that is important, whose target value should be achieved at any efficiency.

Response 9. Thank you for your valuable observation. It is indeed the grafting degree, not efficiency that we calculated. We corrected “efficiency” with “degree” and we added the efficiency % in brackets.

Point 10. Line 314: “16.70 mg ± 0.39 ferulic acid (FA) /g chitosan (approximately 13.7 molar ratio FA:chitosan)”. Do the authors mean that 13.7 molecules of ferulic acid were grafted onto one monomeric unit of chitosan? Maybe it’s a millimolar ratio?

Response 10. We determined the amount of grafted ferulic by the ninhydrin assay as well, in order to have an additional method and we calculated the average between these two methods (lines 505-516, Figure S1). The new molar ratio is 10.78 (considering also the corrected MWt of chitosan). This ratio indicates the number of ferulic acid molecules per molecule of chitosan, considering the chitosan molecular weight.

Point 11. Non-round numbers are striking: “0.432 g of ascorbic acid”, “12.04 mg/mL”, etc. What was the reason for this choice of masses and concentrations? If round numbers of moles (molarities) were used, this should be noted. And sometimes numbers are given with clearly excessive precision (for example, “10.591 ± 0.235” in Table 1). The explanation “titanium (Ti)” (line 795) is redundant—the reader is probably familiar with the Periodic Table.

Response 11. The values mentioned resulted from the protocol that we used to graft the ferulic acid. The values there were also non-round (without any explanation), but reported to 0.25 g chitosan and we considered to leave the same. We removed 12.04 mg/mL and used molar concentration instead. We did not optimize further the parameters; therefore, we wrote exactly the values that resulted from our calculations (extrapolated to 2 g chitosan).

We have rounded the numbers to 2 decimal places in Table 1 (it is average of 3 biological replicates, we added this information under Table 1 and in Materials and Methods). We removed the Ti symbol.

Point 12. A number of abbreviations are defined many times. NDBNC (lines 60 and 155). TGA (lines 200 and 419). SeBNCFa (lines 163, 325, 334, 368, 422, 441, and 667). BNCFa (lines 154, 434, 440, 466, 499, 780, and 815).

The manuscript can be recommended for publication after eliminating the noted shortcomings.

Response 12. We removed all the extra definitions from the main text with the exception of the explanations of the abbreviations from the Figure legends. Thank you for all your valuable suggestions.

Round 2

Reviewer 3 Report

Comments and Suggestions for Authors

The authors took into account almost all of my comments and significantly revised their manuscript. I think it is now suitable for publication, but there are few minor errors to be corrected (no further review required).

Line 201: “acid acetic/acetat de sodiu”—is that in Romanian? An appropriate English term is required.

Eqs (1–3) are well known in polymer science, like the Mark–Houwink–Sakurada one, and can be omitted from the text.

Intrinsic viscosity should be written in square brackets: “[η]”.

Subsection 3.1: Immediately after the value of intrinsic viscosity, the average molecular weight of chitosan should be given.

Line 248: What does “[47]” at the beginning of the line mean?

Such precision (number of significant digits) as in line 1404: “12.449 ± 0.653 mm” should be avoided.

Figure S1: a zero is needed on the x-axis, because the intersection occurs above the 4.8 mark. Also, the dimension along the ordinate axis should be specified: dl/g.

Finally, I would like to commend the expressive figures in the paper.

Author Response

Point 1. Line 201: “acid acetic/acetat de sodiu”—is that in Romanian? An appropriate English term is required.

Response 1.  Thank you for your observation. We corrected it from Romania into English.

Point 2. Eqs (1–3) are well known in polymer science, like the Mark–Houwink–Sakurada one, and can be omitted from the text.

Response 2. We would like to keep these equations, if possible, to help the readers less familiar with this domain understand the paper without too much searching

Point 3. Intrinsic viscosity should be written in square brackets: “[η]”.

Response 3. We corrected.

Point 4. Subsection 3.1: Immediately after the value of intrinsic viscosity, the average molecular weight of chitosan should be given.

Response 4. We added the molecular weight.

Point 5. Line 248: What does “[47]” at the beginning of the line mean?

Response 5. It was a mistake, we deleted it.

Point 6. Such precision (number of significant digits) as in line 1404: “12.449 ± 0.653 mm” should be avoided.

Response 6. We reduced to two digits.

Point 7. Figure S1: a zero is needed on the x-axis, because the intersection occurs above the 4.8 mark. Also, the dimension along the ordinate axis should be specified: dl/g.

Response 7. We updated Figure S1 with the neccessary modifications.

Thank you for all the valuable suggestions.